# Mapping cerebral blood perfusion and its links to multi-scale brain organization across the human lifespan

Asa Farahani[1], Zhen-Qi Liu[1], Eric G. Ceballos[1], Justine Y. Hansen[1], Karl Wennberg[1], Yashar Zeighami[2], Mahsa Dadar[2], Claudine J. Gauthier[3,4], Alain Dagher[1], Bratislav Misic[1]*

**1** Montréal Neurological Institute, McGill University, Montréal, Québec, Canada, **2** Douglas Mental Health Institute, McGill University, Montréal, Québec, Canada, **3** Department of Physics, Concordia University, Montréal, Québec, Canada, **4** Montréal Heart Institute, Montréal, Québec, Canada

* bratislav.misic@mcgill.ca

**Data availability statement:** All code used to perform the analyses are available on GitHub at

## Abstract

Blood perfusion delivers oxygen and nutrients to all cells, making it a fundamental feature of brain organization. How cerebral blood perfusion maps onto micro-, meso- and macro-scale brain structure and function is therefore a key question in neuroscience. Here we analyze pseudo-continuous arterial spin labeling (ASL) data from 1305 healthy individuals in the HCP Lifespan studies (5–22 and 36–100 years) to reconstruct a high-resolution normative cerebral blood perfusion map. At the cellular and molecular level, cerebral blood perfusion co-localizes with granular layer IV, biological pathways for maintenance of cellular relaxation potential and mitochondrial organization, and with neurotransmitter and neuropeptide receptors involved in vasomodulation. At the regional level, blood perfusion aligns with cortical arealization and is greatest in regions with high metabolic demand and resting-state functional hubs. Looking across individuals, blood perfusion is dynamic throughout the lifespan, follows micro-architectural changes in development, and maps onto individual differences in physiological changes in aging. In addition, we find that cortical atrophy in multiple neurodegenerative diseases (late-onset Alzheimer's disease, TDP-43C, and dementia with Lewy bodies) is most pronounced in regions with lower perfusion, highlighting the utility of perfusion topography as an indicator of transdiagnostic vulnerability. Finally, we show that ASL-derived perfusion can be used to delineate arterial territories in a data-driven manner, providing insights into how the vascular system is linked to human brain function. Collectively, this work highlights how cerebral blood perfusion is central to, and interlinked with, multiple structural and functional systems in the brain.

## Introduction

The brain is a complex structure spanning multiple spatial scales [1,2]. At the cellular level, neurons with a rich array of morphologies and receptor profiles assemble into layered and increasingly poly-functional neural circuits [3–7]. The propagation of electrical impulses

https://github.com/netneurolab/Farahani_
Blood_Perfusion/ and on Zenodo at
https://zenodo.org/records/15708107 (DOI:
10.5281/zenodo.15708107). Arterial spin
labeling (ASL), functional MRI, and structural
MRI are accessible through the Human
Connectome Project–Development (HCPD;
https://www.humanconnectome.org/study/
hcp-lifespan-development/) and the Human
Connectome Project–Aging (HCP-A;
https://www.humanconnectome.org/study/
hcp-lifespan-aging/).

**Funding:** AF acknowledges support from the
Molson Foundation. ZQL acknowledges support
from the Fonds de Recherche du Québec –
Nature et Technologies (FRQNT). JYH
acknowledges support from the Helmholtz
International BigBrain Analytics & Learning
Laboratory, the Natural Sciences and
Engineering Research Council of Canada
(NSERC) and the Neuro-Irv and Helga Cooper
Foundation. EGC acknowledges support from
the Molson Foundation and FRQNT. BM
acknowledges support from the NSERC
(RGPIN-2017-04265), Canadian Institutes of
Health Research (CIHR) (PJT-180439), Brain
Canada Foundation Future Leaders Fund, the
Canada Research Chairs Program
(CRC-2022-00169), the Michael J. Fox
Foundation (MJFF-021133) and the Healthy
Brains for Healthy Lives initiative. The funders
have no role in study design, data collection and
analysis, decision to publish or preparation of
the paper.

**Competing interests:** The authors have
declared that no competing interests exist.

across this network manifests as patterned neural activity [8–11]. Advances in imaging and recording technologies, coupled with data sharing initiatives, provide increasingly detailed maps of brain structure and function at multiple scales of description, including gene expression [12–15], neurotransmitter receptors [16–21], cell types [22–25], laminar differentiation [26–29], metabolism [30,31], and neurophysiological activity [32–37].

Central to the development and function of the brain is its blood supply [38–42]. Blood perfusion delivers oxygen and nutrients for all cells, making it a fundamental feature of brain organization [43–45]. Despite their intertwined nature, the study of the brain and its vasculature have diverged as two separate scientific domains [46,47]. However, there exists shared anatomical patterning between nervous and vascular networks throughout the whole body [48–50]. Blood vessels and neurons direct their outgrowth using shared molecular mechanisms, ultimately resulting in similarly patterned networks [51–55]. In other words, metabolic need and metabolic support in the brain are inseparable and should be studied in tandem. How blood perfusion maps onto canonical features of brain structure and function remains a key question in the field.

Indeed, numerous technologies have been developed to quantify blood perfusion, defined as volume of arterial blood delivered to a unit mass of brain tissue per unit time. Initial efforts to quantify cerebral blood flow (CBF) used nitrous oxide inhalation for blood flow quantization [56,57], while more recent variants of the technique use single photon emission computed tomography (SPECT) [58,59], positron emission tomography (PET) [60,61] and magnetic resonance imaging (MRI) with contrast agents [62–65]. More recently, pseudo-continuous arterial spin labeling (ASL) permits accurate measurements that are non-invasive and have greater spatial resolution [61,66–72]. In other words, parallel advances in blood perfusion imaging present the opportunity to systematically compare maps of brain structure and function to maps of blood perfusion.

Here we comprehensively characterize blood perfusion in the brain at multiple levels of description. We first investigate how blood perfusion co-localizes with micro- and meso-scale features, including neurotransmitter signaling, cell types and cortical layers. At the macro-scale, we investigate how blood perfusion maps onto regional differences in metabolism and hub connectivity. Across individuals, we further investigate how blood perfusion changes through the lifespan and in multiple neurodegenerative diseases. Finally, we show that inter-regional coordination of blood perfusion can be used to map arterial territories. Collectively, these experiments establish blood perfusion as an important biological annotation for future multi-scale exploration of brain anatomy and function.

## Results

We use pseudo-continuous arterial spin labeling (ASL) magnetic resonance imaging (MRI) as part of the Human Connectome Project–Development (HCP-D; https://www.humanconnectome.org/study/hcp-lifespan-development/) and the Human Connectome Project–Aging (HCP-A; https://www.humanconnectome.org/study/hcp-lifespan-aging/) to derive quantitative maps of brain blood perfusion [73–75]. In subsequent analyses, data from 1305 participants (5–22 and 36–100 years; 718 female) are included (Fig 1). See *Methods* for details about participants' demographic information, data acquisition, and preprocessing.

### Mapping blood perfusion across individuals and brain regions

We start by deriving a normative map of cerebral blood perfusion using arterial spin labeling (ASL) imaging technique. We first estimate blood perfusion (defined as the volume of blood delivered to a unit of tissue per unit of time) in both cortex (vertex-wise) and subcortex

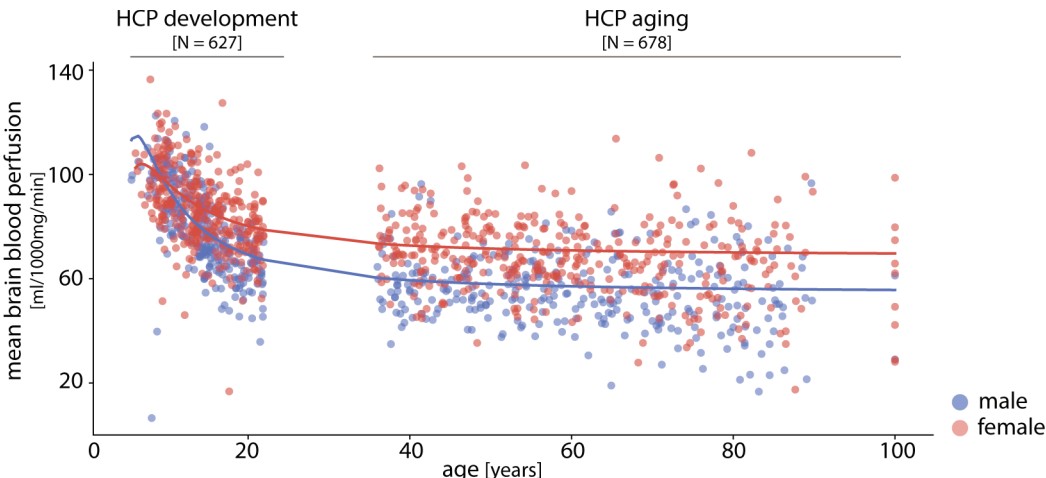

**Fig 1. Mapping cerebral blood perfusion across participants.** Cerebral blood perfusion is estimated using arterial spin labeling (ASL) from the HCP Lifespan studies, including 627 participants (337 females) from the HCP-D (development) and 678 participants (381 females) from the HCP-A (aging). Each dot corresponds to a participant's mean brain (gray-ordinates) blood perfusion level (male: blue, female: red) (See S1 Fig for the mean perfusion trajectory within white matter mask). Sex-stratified generalized additive models for location, scale and shape (GAMLSS) [76] are used to model age-related changes in blood perfusion.

(voxel-wise) for each participant. Numerous reports have noted greater perfusion in females compared to males [77–85]. Here we stratify participants according to biological sex and observe the same phenomenon (S2 Fig). S2B Fig shows the correlation between perfusion values across the two groups ($r = 0.99$). We confirm greater perfusion in female brain ($t = 9.27$, $p_{\text{two-sided}} = 7.21 \times 10^{-20}$; S2B Fig). In addition, we observe lower perfusion in subcortex compared to cortex in both groups (male: $t = 22.96$, $p_{\text{two-sided}} = 1.25 \times 10^{-96}$; female: $t = 30.65$, $p_{\text{two-sided}} = 3.96 \times 10^{-159}$; S3 Fig), consistent with previous reports [85].

We next sought to generate a common blood perfusion map that is representative of all participants in the sample. To do so, we z-score blood perfusion maps of individual participants and concatenate them into a single data matrix. The standardization at the participant level mitigates perfusion level offsets attributed to biological sex differences. We next apply principal component analysis (PCA) dimensionality reduction to the data matrix. The first principal component accounts for 50.71% of the variance in blood perfusion across the sample (the second component accounts for 2.35% of the variance). This first component reflects the main regional pattern of blood perfusion shared across participants (Fig 2A and 2B). Hereafter, we refer to this map as the "perfusion score map". The corresponding participant-level loadings for the first principal component are shown in S4A Fig. Encouragingly, the ASL-derived perfusion score map is significantly correlated with PET-estimated cerebral blood flow ($r = 0.63$, $p_{\text{spin}} = 9.99 \times 10^{-4}$; see S5 Fig).

For completeness, in S6 Fig we show an alternative method for deriving a group-representative map by regressing out linear and non-linear age and sex effects from individual perfusion maps prior to applying PCA. The perfusion score map from this analysis (S6A Fig) correlates with the original map shown in Fig 2A and 2B ($r = 0.93$, S6C Fig), confirming that both methods yield similar spatial patterns of blood perfusion and that the age and sex effects are not central to this component. Notably, individual-level ASL maps have low signal-to-noise ratios; using PCA to construct a representative map for blood perfusion has the advantage of relying on shared variance across the dataset and being less influenced by

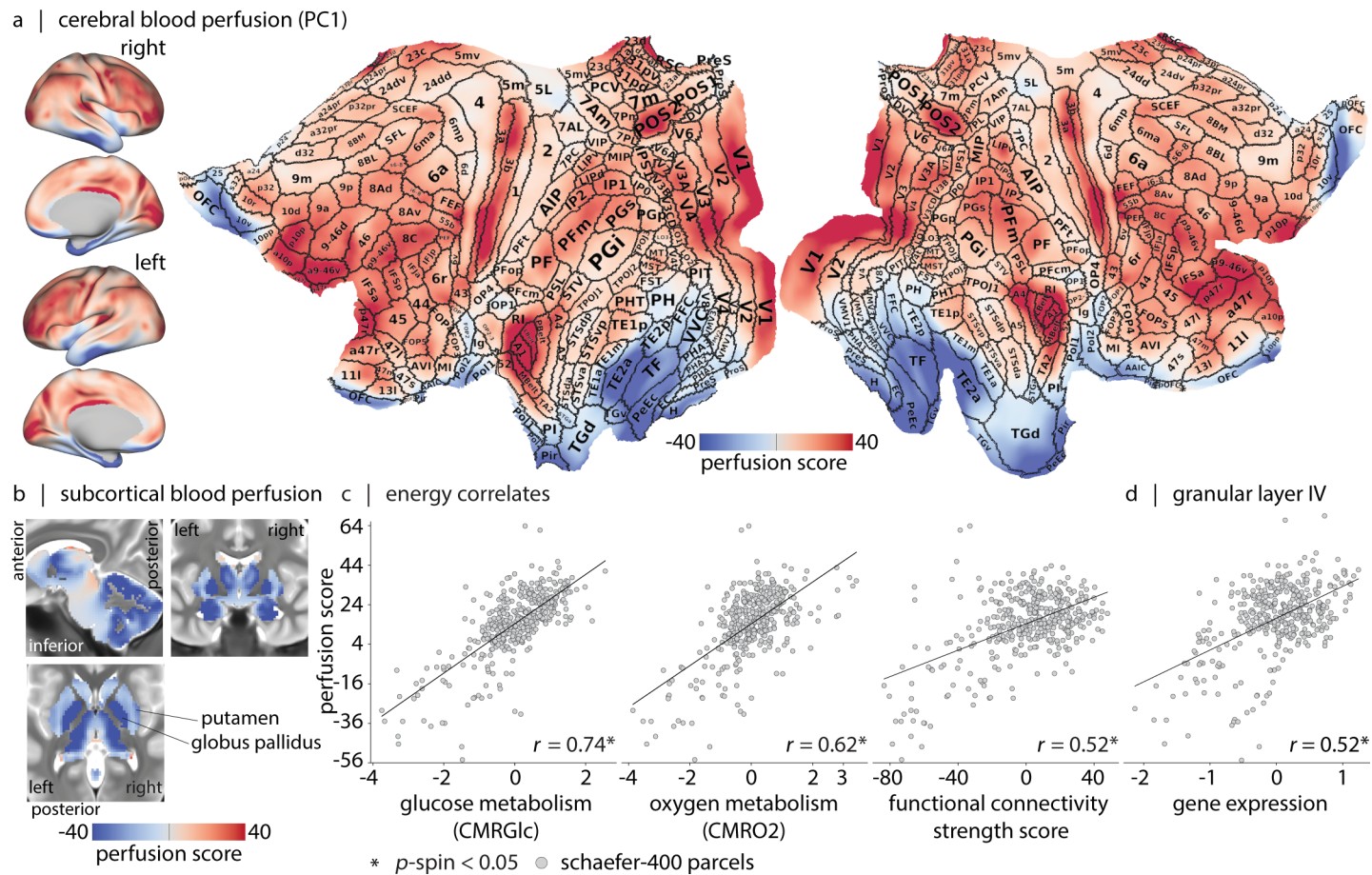

**Fig 2. Areal organization of blood perfusion.** (a) Normative blood perfusion map on lateral and medial views of the inflated and 2D flat cortical surfaces (fsLR). Borders and areal names of the multi-modal Glasser parcellation are overlaid on the flat surface [86]. The figure highlights the non-uniform distribution of perfusion scores across the cortex, with areas that have sharp gradient compared to their underlying perfusion score levels (e.g., LIPv and MT/MST) (see S1 and S2 Tables). (b) Subcortical perfusion is shown on a T2-weighted group average template. Quantification of perfusion scores according to the Tian-S4 subcortical parcellation is shown in S7 Fig [87]. A complete list of subcortical parcels is provided in S3 Table. (c) Correlations between perfusion ($y$-axis) and measurements of energy consumption ($x$-axis). Left: Correlation with glucose consumption (CMR$_{Glc}$; $r = 0.74$, $p_{spin} = 9.99 \times 10^{-4}$) [30]. Middle: Correlation with oxygen consumption (CMR$_{O2}$; $r = 0.62$, $p_{spin} = 9.99 \times 10^{-4}$) [30]. Right: Correlation with resting-state functional MRI connectivity strength (a measure of "hubness") ($r = 0.52$, $p_{spin} = 2.99 \times 10^{-3}$; see *Methods* for more details). (d) The transcriptomic signature of cortical layer IV [88] is positively correlated with perfusion ($r = 0.52$, $p_{spin} = 2.99 \times 10^{-3}$). We do not find correspondence between blood perfusion and transcriptomic signature of supragranular (I–III) and infragranular (V–VI) cortical layers (S8 Fig).

outliers in the data compared to simple averaging. Nevertheless, given the large sample size in this study ($N = 1305$), the perfusion score map and the mean perfusion map exhibit highly similar spatial patterns (compare Figs 2A, 2B, and S2A).

## Areal organization of blood perfusion

We explore how perfusion co-localizes with canonical features of brain structure and function. Fig 2A and 2B suggest that cerebral blood perfusion follows areal boundaries (e.g., V1–V2, 3b–1, 3a–4, putamen–globus pallidus). Sorting regions according to blood perfusion reveals the greatest scores in sensory areas (see S2 Table), particularly in early-auditory areas including primary auditory cortex (A1), lateral belt complex (LBelt), medial belt complex (MBelt), and Para-belt complex (PBelt) [89], early somatosensory areas including 3b and 3a,

and early visual areas including the V1 strip. We also observe high perfusion in dorsolateral frontal cortex and heavily myelinated area POS2 (Fig 2A) [86]. In the subcortex, we observe the greatest perfusion in posterior thalamus (DAm) and tail of caudate, and the lowest perfusion in globus pallidus and amygdala (Fig 2B, see S7 Fig for more details). A common feature of most of the high-perfusion structures is prominent sensory input. Collectively, these results show that resource allocation in the brain, as reflected by blood perfusion, is highly heterogeneous.

As the first step towards understanding how perfusion corresponds with other features of brain organization, we ask whether blood perfusion—the primary system for resource allocation—reflects energy consumption of the underlying tissue (Fig 2C). Here we assess whether cortical blood perfusion maps onto glucose and oxygen metabolism [90]. Cerebral metabolic rate of glucose ($CMR_{Glc}$) and cerebral metabolic rate of oxygen ($CMR_{O2}$) are estimated using positron emission tomography (PET) in 33 healthy adults (see *Methods*). Specifically, $CMR_{Glc}$ is derived using [$^{18}$F]-labeled fluorodeoxyglucose (FDG) and $CMR_{O2}$ is derived using administration of [$^{15}$O]-labeled oxygen [30]. The blood perfusion score map is correlated with both $CMR_{Glc}$ ($r = 0.74$, $p_{spin} = 9.99 \times 10^{-4}$) and $CMR_{O2}$ ($r = 0.62$, $p_{spin} = 9.99 \times 10^{-4}$). Greater blood perfusion is also associated with greater functional connectivity strength at both regional (Figs 2C and S4B) and whole brain levels (S4C Fig) [91].

Given that perfusion is regionally heterogeneous, we next test whether perfusion patterns co-localize with regional differences in laminar differentiation. Specifically, we test whether perfusion is spatially correlated with granular layer IV, the primary input layer of the cortex. Fig 2D shows the spatial correlation between ASL-estimated perfusion and the average expression map of five genes preferentially expressed in this layer [88,92]. The observed positive correlation with perfusion ($r = 0.52$, $p_{spin} = 2.99 \times 10^{-3}$) is consistent with previous imaging studies reporting denser regional vasculature of primary sensory areas and specifically cortical layer IV in human [93–95], macaque monkey [96–98], cat [99], rat [100] and mouse brain [101–103]; suggesting that the early stages of feed-forwarding processing are highly resource demanding. Collectively, these results build on animal and region-of-interest studies, showing that perfusion and areal differentiation are closely intertwined at the whole-brain level. In the following subsection, we explore the molecular and cellular infrastructure that shape this vascular resource allocation.

## Blood perfusion co-localizes with molecular and cellular features

Given that perfusion co-localizes with macro- and meso-scale features, we next sought to identify micro-scale (1) molecular pathways, (2) cell types and (3) receptors that contribute to regional differences in perfusion. First, we cross-reference the perfusion score map with microarray gene expression from the Allen Human Brain Atlas (AHBA) [13]. We submit the resulting gene list to gene category enrichment analysis (GCEA) to isolate gene ontology (GO) categories in which the constituent genes are significantly more correlated with the perfusion score map than a population of random score maps with preserved spatial auto-correlation [104,105] (see *Methods*). The results indicate that the perfusion score map co-localizes with GO categories directly related to: the vascular system ("vasculogenesis"); brain metabolism ("mitochondrial organization", "cellular response to glucose stimulus", "regulation of insulin secretion", "lipid transport"); and the synaptic activity ("sodium ion transmembrane transport", "potassium ion import across plasma membrane", "potassium ion transmembrane transport", "regulation of ion transmembrane transport") (Fig 3A; see full list in S4 Table).

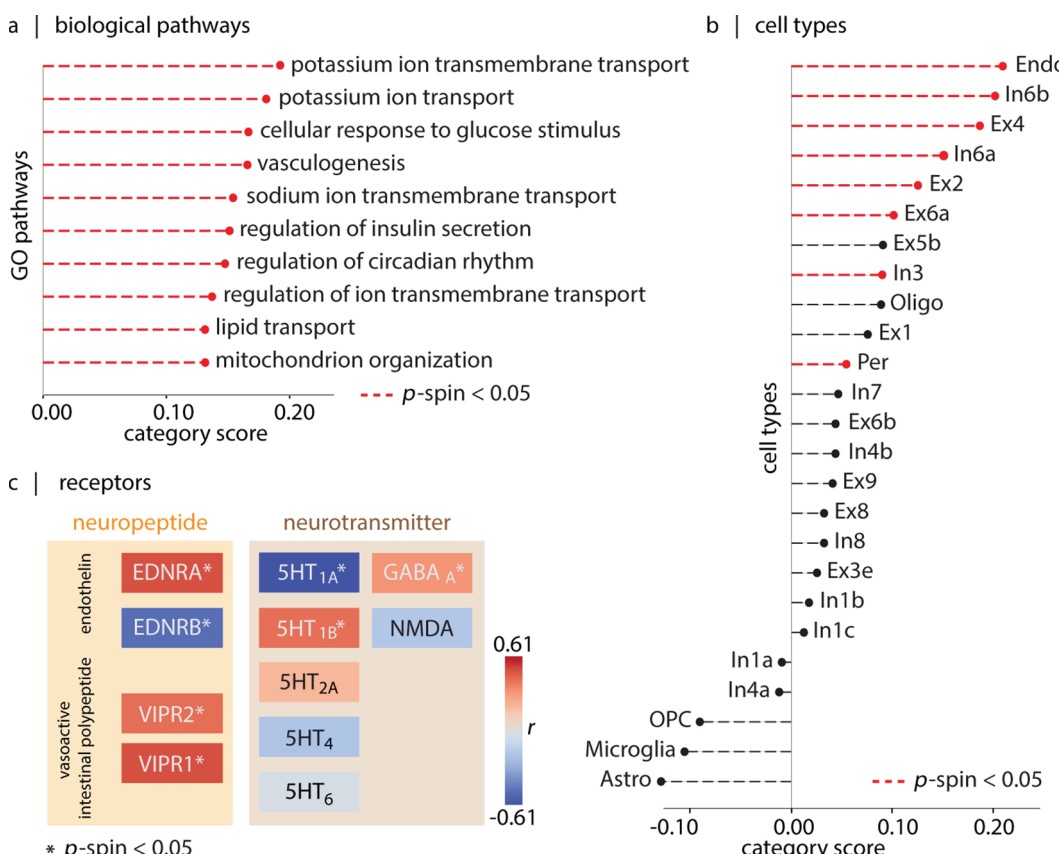

**Fig 3. Blood perfusion co-localizes with molecular and cellular features.** (a) The top ten GO biological processes linked with gene sets correlated with perfusion [104]. Terms are sorted based on their associated category-scores. For a complete list of genes associated with a biological pathway, category-scores and statistics, see S4 Table. (b) Cell type enrichment analysis of the cortical perfusion score map [23]. For list of genes associated with a cell type, category scores and statistics, see S5 Table. Red lines indicate statistically significant associations (after FDR-correction). (c) Comparison of perfusion with maps of neurotransmitter receptors (from PET imaging) and neuropeptide receptors (from transcriptomics) [18,106]. Colors indicate directions of association (red: positive, blue: negative) and asterisks indicate significant associations. See S9 Fig for results obtained using a multivariate model with dominance analysis. Methodological details about each neurotransmitter receptor PET map are provided in S6 Table.

Second, we find that the perfusion score map is enriched for transcriptomic markers of endothelial cells and pericytes [23,105], both of which are principal constituents of the neurovascular unit [107,108]. The score map also co-localizes with the In6a, In6b, Ex4, Ex2, Ex6a and In3 cell types (Fig 3B) [23,105]. In6a and In6b cell types are fast-spiking parvalbumin-positive GABAergic neurons (In6a: PVALB$^+$CA8$^+$ concentrated around layer IV; In6b: more peripheral PVALB$^+$TAC1$^+$ [23]). These fast-spiking neurons are thought to incur greater energetic cost [109–112]. Interestingly, Ex4 and Ex2 are excitatory cell markers of layer IV [23].

Third, we investigate the relationship between cerebral perfusion and two molecular signaling systems: neurotransmitter receptors and neuropeptide receptors (Fig 3C). Briefly, neurotransmitter receptor maps are recovered from a normative PET atlas [18,113], while neuropeptide maps are recovered from the AHBA ([106]; see *Methods*). We restrict the analysis only to systems that have previously been linked to vascular function, including serotonin receptors [114–118], GABA, NMDA, endothelin (ENDR), and vasoactive intestinal

peptide (VIPR) receptors. Among the serotonin receptors, $5HT_{1A}$ ($r = -0.61$, FDR-corrected $p_{spin} = 3.49 \times 10^{-3}$) and $5HT_{1B}$ ($r = 0.42$, FDR-corrected $p_{spin} = 3.49 \times 10^{-3}$) show association with cerebral blood perfusion pattern, while $5HT_{2A}$ ($r = 0.20$), $5HT_4$ ($r = -0.20$) and $5HT_6$ ($r = -0.03$) do not show significant associations. $GABA_A$ also aligns with blood perfusion pattern ($r = 0.32$; FDR-corrected $p_{spin} = 4.66 \times 10^{-3}$); in contrast NMDA does not show association with blood perfusion ($r = -0.18$). All studied neuropeptide receptors correlate with the blood perfusion pattern (EDNRA, $r = 0.49$; EDNRB, $r = -0.51$; VIPR1, $r = 0.42$; VIPR2, $r = 0.49$; all with FDR-corrected $p_{spin} = 9.99 \times 10^{-3}$). Collectively, this enrichment analysis highlights a convergence of vascular organization with molecular mechanisms involved in cellular development, activity and metabolism.

## Blood perfusion across development

So far, we characterized normative blood perfusion and situated it among multiple structural and functional features of brain organization. However, cerebral blood perfusion is dynamic and changes throughout the lifespan [85,119–123]. Here we reconstruct normative trajectories of blood perfusion using the HCP datasets, which cover a wide age range (5–22 and 36–100 years) with high spatial resolution (2.5 mm³ voxels) [85]. Figs 1 and 4A show whole-brain blood perfusion trajectories stratified by biological sex. The overall pattern reveals age-related reduction in perfusion, with a steeper decrease during development compared to aging. Sex differences in blood perfusion begin to emerge during development, become more pronounced around puberty, and persist into adulthood (Fig 4A, S7 Table, and S10 Fig) [122].

We then explore which brain regions experience the greatest developmental changes in blood perfusion. Fig 4B shows the regional linear age-effect in HCP-D, with greater age-related reductions mainly in association cortex. To confirm this intuition, we stratify regions according to their affiliation with unimodal or transmodal intrinsic networks [124,126], and compute the mean age-effect in each category. Fig 4C shows that age-related reductions in perfusion during development are significantly greater in transmodal compared to unimodal cortex ($t = 12.97$, $p = 2.03 \times 10^{-30}$). These findings are consistent with the notion that perfusion tracks the extensive micro-architectural remodeling during development. Namely, there is an initial increase in perfusion in early childhood (0–7 years) in response to increased synaptogenesis and myelination [127–133]. This phase is followed by a decline in perfusion during late childhood and adolescence as synaptic pruning progresses [134]. Synaptogenesis and synaptic pruning occur earlier and more rapidly in unimodal cortex [129] but stabilize by the age of the HCP-D cohort (5+ years). In contrast, transmodal cortex, characterized by the greatest overproduction and slowest elimination of dendritic spines [135,136], continues to undergo developmental changes during this age range (Fig 4B and 4C).

For completeness, we also used GAMLSS models to build sex-stratified trajectories of perfusion at both whole-brain (grayordinates) and parcel-level (Schaefer-400 parcels) scales (S14 and S15 Figs). GAM models can capture non-linear age effects and are of interest in lifespan [76,137,138] and specifically developmental neuroimaging studies [139]. Using this approach, we replicate the greater age-related perfusion changes in transmodal regions during development (5–22 years).

To test the link between developmental changes in perfusion and grey matter tissue, we correlate age-related changes in cortical thickness (estimated using T1- and T2-weighted MRI) with linear age-related changes in perfusion. We find that decreases in cortical perfusion with age are accompanied by reductions in cortical thickness, highlighting the close link between grey matter and vascular development ($r = 0.33$, $p_{spin} = 3.99 \times 10^{-3}$; Fig 4D).

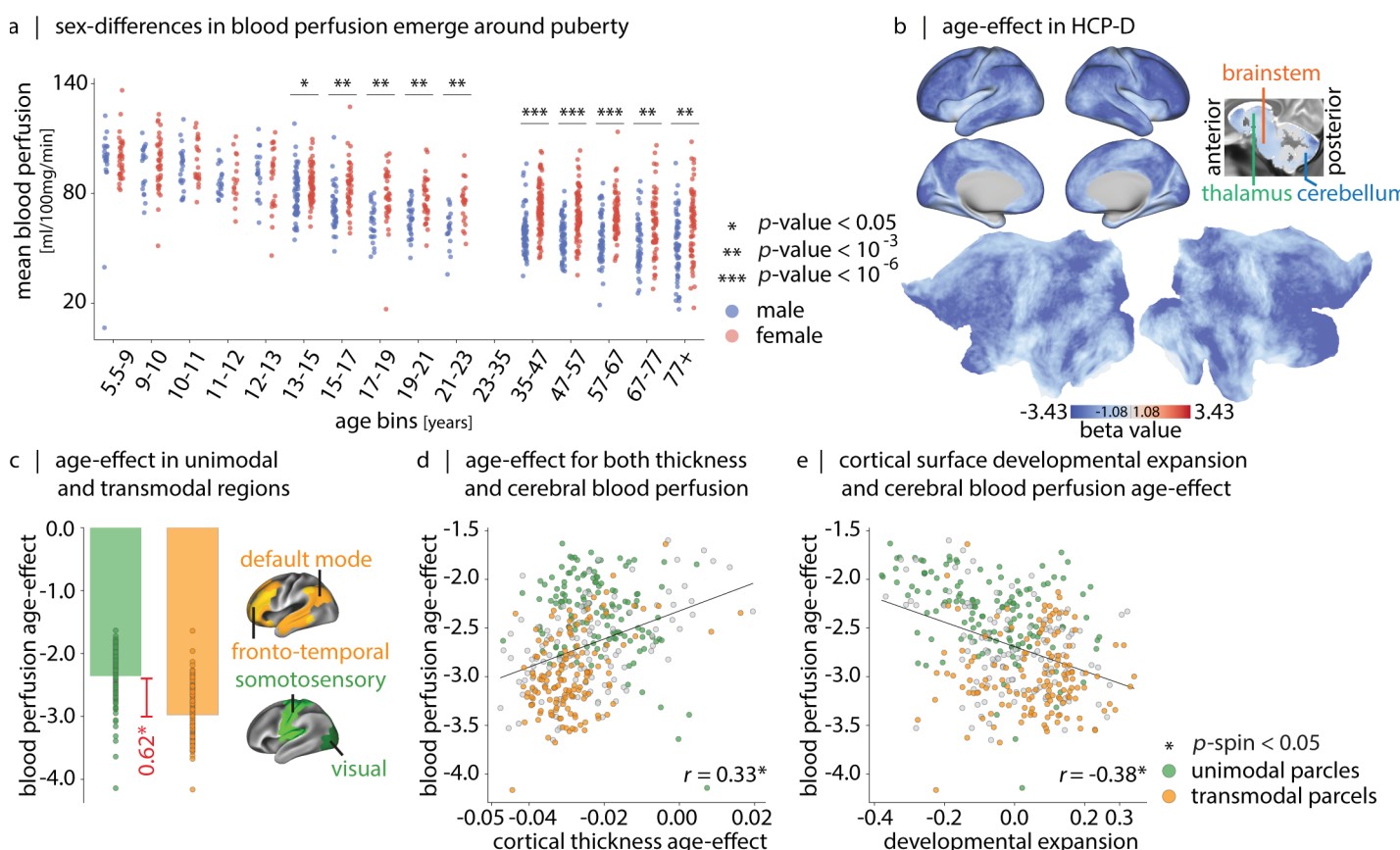

**Fig 4. Blood perfusion across development.** (a) Whole-brain perfusion ($y$-axis), stratified by age ($x$-axis) and biological sex (male: blue, female: red). S7 Table shows statistical comparisons between males and females in each age bin. (b) Age-effect perfusion map estimated using data from 627 participants in the HCP-Development dataset (5–22 years). Linear age coefficients ($\beta_1$) are derived from a linear model ($\mathrm{CBF}_i = \beta_0 + \beta_1 \times \mathrm{age} + \beta_2 \times \mathrm{sex}$), applied at each vertex/voxel $i$. Subcortical age-effect values are shown in S11A Fig. Sex-effect and coefficient significance maps are provided in S12 Fig. S13 Fig shows vertex/voxel-wise Spearman correlation maps of blood perfusion and age, stratified by biological sex. See S14 and S15 Figs for sex-stratified non-linear modeling of perfusion changes. (c) Mean age-effect on cerebral perfusion, stratified into unimodal and transmodal cortex (unimodal: visual and somatomotor networks, shown in green; transmodal: frontoparietal and default mode networks, shown in yellow; canonical intrinsic functional networks from Yeo et al. [124]). The age-effect difference ($\Delta\beta_1 = |0.62|$) is significant using both $t$-tests ($t = 12.97$, $p = 2.03 \times 10^{-30}$) and non-parametric spin tests ($p_{\mathrm{spin}} = 9.99 \times 10^{-4}$). (d) Correlation between cerebral blood perfusion changes ($y$-axis) and cortical thickness changes ($x$-axis) ($r = 0.33$, $p_{\mathrm{spin}} = 3.99 \times 10^{-3}$). (e) Correlation between cerebral blood perfusion changes ($y$-axis) and developmental cortical expansion ($x$-axis) ($r = -0.38$, $p_{\mathrm{spin}} = 9.99 \times 10^{-4}$) [125]. In (c), (d) and (e) maps are parcellated according to Schaefer-400 atlas [126].

Pursuing this idea further, we identify a negative association between normative cortical surface expansion during development (identified by Hill et al. [125]) and the perfusion age-effect map ($r = -0.38$, $p_{\mathrm{spin}} = 9.99 \times 10^{-4}$; Fig 4E). Collectively, these results highlight the dynamic coupling between cerebral structure and blood perfusion.

## Blood perfusion across typical aging

As a complement, we next investigate cerebral blood perfusion changes in typical aging. Fig 5A shows the linear effect of age on regional blood perfusion, with age-related reductions observed in most brain regions (see S11B Fig for subcortical effect). Notably, regions with the greatest age-related reductions appear to be situated along the putative vascular border-zones (so called "watershed" regions), defined as regions most distant from the main cerebral arteries [140,141]. To confirm this intuition, we derive maps of border-zone regions using two

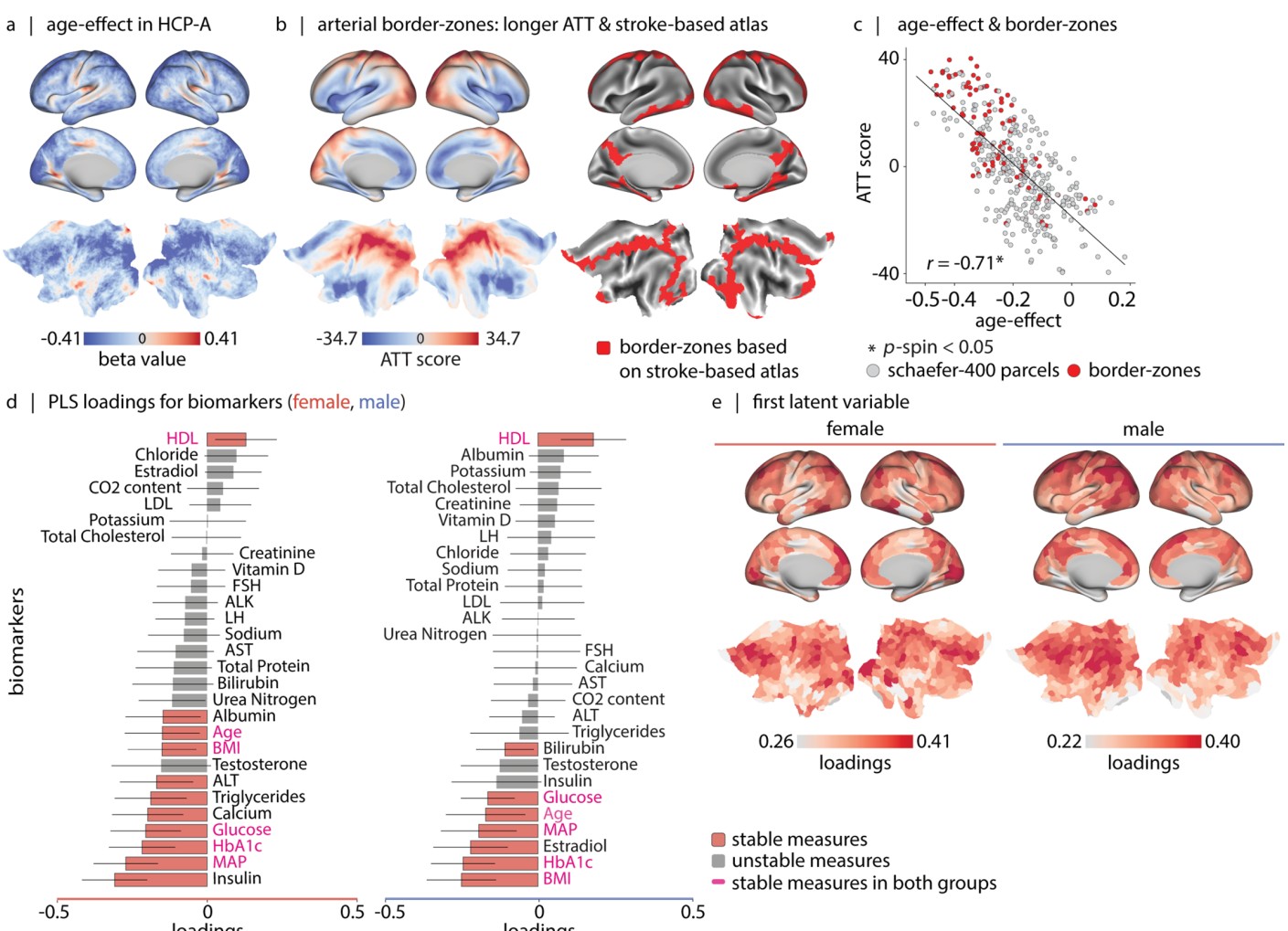

**Fig 5. Blood perfusion across aging.** (a) Age-effect perfusion map estimated using data from 678 participants in the HCP-Aging dataset (36–100 years). Linear age coefficients ($\beta_1$) are derived from a linear model (CBF$_i$ = $\beta_0$ + $\beta_1$ × age + $\beta_2$ × sex), applied at each vertex/voxel $i$. Subcortical age-effect values are shown in S11B Fig. Sex-effect and coefficient significance maps are provided in S16 Fig. S17 Fig shows vertex/voxel-wise Spearman correlation maps of blood perfusion and age, stratified by biological sex. See S18 Fig for sex-stratified non-linear modeling of perfusion changes. (b) The age-effect pattern aligns with arterial border-zone cortical regions. The border-zone areas are described based on: (1) Left: First principal component of estimated arterial transit time maps (greater values correspond to regions with later arterial transit time; see S19 Fig for analysis details). In the figure, "ATT" stands for arterial transit time. (2) Right: Intersections of major cerebral arterial territories derived from a stroke-based atlas [142,143]. (c) Correlation between cerebral blood perfusion age-related changes ($x$-axis) and arterial transit time scores ($y$-axis) ($r$ = –0.71, $p_{spin}$ = 9.99 × 10$^{-4}$). Red dots represent border-zone regions defined according to (2). (d) Using partial least squares (PLS) analysis, we find a significant latent variable that accounts for 82.0%, and 90.6% of the covariance between cortical blood perfusion and biomarkers (S20 Fig) in male and female participants (in both cases: $p$ = 9.99 × 10$^{-4}$). The bar plot visualizes the contribution of individual biomarkers to the first latent variable. The significance of each biomarker's contribution to the overall pattern is assessed by bootstrap resampling (1 000 bootstraps; see S21 Fig). (e) The brain loadings of the first latent variable are shown for both groups. These maps correlate with the arterial transit time score map shown in (b) (male: $r$ = 0.47, $p_{spin}$ = 0.027, female: $r$ = 0.52, $p_{spin}$ = 9.99 × 10$^{-4}$). Brain maps in (a), (b) and (e) are shown on the inflated and 2D flat cortical surfaces (fsLR); maps in (b)-right and (e) are parcellated according to the Schaefer-400 functional atlas [126]. Results of PLS analysis after regressing out the linear effect of age and sex from the perfusion maps and the biomarkers are presented in S22 Fig.

complementary methods (Fig 5B): (1) using ASL-estimated arrival transit times (ATT), which identifies regions that are positioned distal to the major arteries, and (2) using an independent atlas derived from a stroke dataset [142,143]. Fig 5C confirms that reductions in perfusion are greatest in arterial border-zones ($r$ = –0.71, $p_{spin}$ = 9.99 × 10$^{-4}$).

We further ask how physiological attributes contribute to the spatial pattern of age-related decline in cerebral blood perfusion. We use partial least squares (PLS) analysis to identify a multivariate mapping between regional cerebral blood perfusion levels and 28 biomarkers per biological sex group [144,145]. These measures include blood laboratory results, age, mean arterial pressure (MAP), and body mass index (BMI). Blood laboratory parameters encompass total protein, glucose, insulin, hemoglobin A1c (HbA1c), triglycerides, low-density lipoprotein (LDL), high-density lipoprotein (HDL), total cholesterol, albumin, bilirubin, creatinine, urea, chloride, sodium, potassium, calcium, vitamin D, and $CO_2$ content, liver metabolic enzymes (ALT, AST, ALP), and hormonal measures (estradiol, testosterone, LH, and FSH) (S20 Fig shows the raw measurement values versus participants' age). The first latent variable accounts for 82.0%, and 90.6% of the shared covariance between cerebral blood perfusion and biomarkers in male and female groups, respectively. Statistical significance is assessed by permutation tests (for both groups: $p = 9.99 \times 10^{-4}$, 1 000 repetitions), and the stability of loadings is assessed by bootstrap resampling (1 000 repetitions; see *Methods*). The biomarkers contributing to this latent variable are shown in Fig 5D; stable biomarkers across both groups include higher HDL level, and lower MAP, BMI, age, glucose and HbA1c values. This pattern of biomarkers is associated with greater cerebral blood flow in the cortex, specifically within border-zone regions (Fig 5E, see *Methods* for details of cross-validation and generalizability of PLS results). Collectively, these results show that reductions in cerebral perfusion during typical aging are highly organized and pronounced in arterial border-zones, and are associated with multiple physiological indicators of cerebrovascular health.

## Cerebral blood perfusion and susceptibility to neurodegeneration

So far, we focused on cartography of cerebral blood perfusion with respect to the underlying brain tissue, and explored how perfusion changes during development and asymptomatic aging. In this section, we elaborate on how the non-uniform distribution of cerebral blood perfusion contributes to the biological vulnerability of underlying brain tissue in pathological conditions. The presence of vascular impairments is well-documented in neurodegenerative diseases [146–153], which can impair the clearance of emboli [154] and neurotoxins [155], diminish transport of energy substrates to the underlying tissue and exacerbate neurodegeneration. Regions with inherently lower blood supply and those located in border-zone areas are particularly prone to emboli build-up [156–158].

Here we cross-correlate atrophy patterns of eight confirmed pathological conditions [159] with the cerebral blood flow score map (Fig 6A). The assessed conditions include early-onset Alzheimer's disease (EOAD), late-onset Alzheimer's disease (LOAD), presenilin 1 (PS1) mutation carriers, three-repeat tauopathy (3Rtau), four-repeat tauopathy (4Rtau), frontotemporal lobar degeneration with TDP-43 type A pathology (TDP-43A), frontotemporal lobar degeneration with TDP-43 type C pathology (TDP-43C), and dementia with Lewy bodies (DLB) [159]. Notably, atrophy in LOAD ($r = -0.43$; FDR-corrected, $p_{spin} = 3.99 \times 10^{-3}$), TDP-43C pathology ($r = -0.61$; FDR-corrected, $p_{spin} = 3.99 \times 10^{-3}$), and DLB pathology ($r = -0.37$; FDR-corrected, $p_{spin} = 5.32 \times 10^{-3}$) happens in areas with inherently lower blood perfusion (Fig 6B–6D). Atrophy is estimated using voxel-based morphometry (VBM) applied to the T1-weighted images (see *Methods*). Our finding suggests that brain regions with low perfusion are particularly susceptible to late-onset AD, DLB and TDP-43C pathologies.

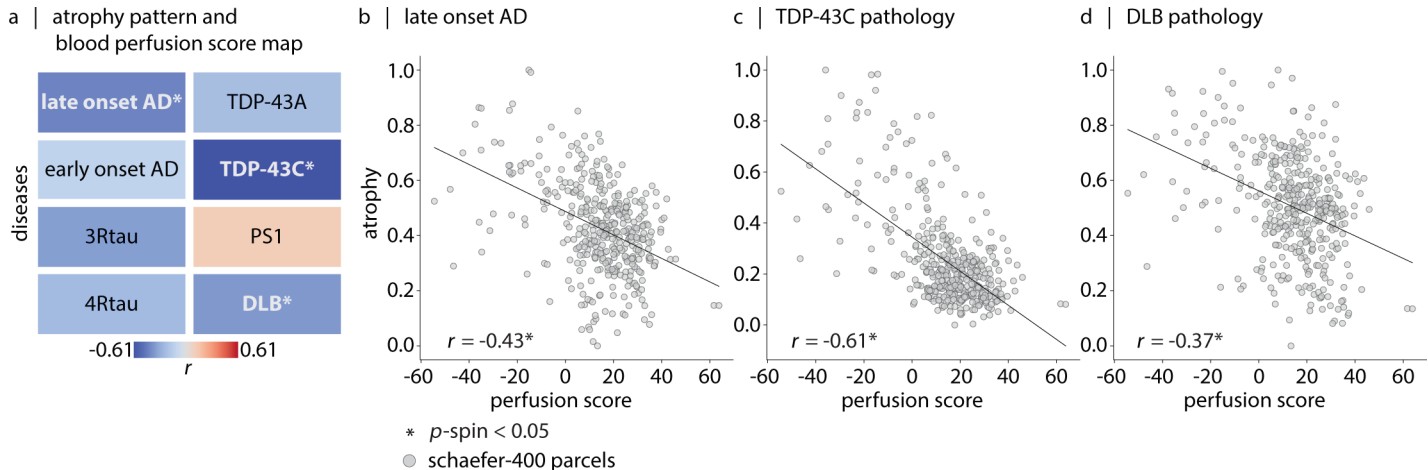

**Fig 6. Cerebral blood perfusion and susceptibility to neurodegeneration.** (a) Comparison of perfusion with eight pathology-confirmed disease atrophy maps (voxel-based morphometry; VBM data from Harper et al. [159]), both parcellated using the Schaefer-400 functional atlas [126]. Colors indicate directions of association (red: positive, blue: negative) and asterisks indicate significant associations. (b) Late-onset Alzheimer's disease ($r = -0.43$; FDR-corrected, $p_{spin} = 3.99 \times 10^{-3}$), (c) TDP-43C ($r = -0.61$; FDR-corrected, $p_{spin} = 3.99 \times 10^{-3}$), (d) and dementia with Lewy bodies ($r = -0.37$; FDR-corrected, $p_{spin} = 5.32 \times 10^{-3}$) show significant negative correlations with the perfusion map, indicating that these disease-specific atrophy patterns co-localize with areas of lower cerebral perfusion.

## Coordinated patterns of blood perfusion outline vascular territories

As a final step, we seek to investigate how blood perfusion is coordinated among multiple brain regions, and the extent to which this network organization can be used to recover the underlying arterial territories. The brain receives blood primarily from two main arterial systems: the internal carotid arteries and the vertebral arteries. The internal carotid arteries give rise to the anterior and middle cerebral arteries and form the anterior circulation of the brain. The vertebral arteries give rise to the basilar and posterior cerebral arteries and form the posterior circulation [160].

To derive arterial territories, we use ASL-derived blood perfusion maps to generate a blood perfusion covariance matrix across participants. This involves $z$-scoring each participant's ASL map followed by $z$-scoring the data across all participants for each vertex/voxel. We then calculate the pairwise correlations between regions to reveal the network structure. This network represents the extent to which blood perfusion in one region covaries with other regions across participants [161,162], and may potentially reflect the underlying vascular anatomy [163].

Here we derive the first two gradients of the perfusion covariance matrix using diffusion map embedding (Fig 7A), and compare them to an independent atlas of arterial territories derived using stroke data (Fig 7B) [142,143]. The first gradient separates the territories of posterior and anterior brain circulations (Fig 7A, top). The second gradient highlights a smooth transition from anterior brain circulation to border-zone brain regions (Fig 7A, bottom). S23 Fig shows these gradients in subcortex, revealing that brainstem, cerebellum, hippocampus, and posterior thalamus are supported by posterior circulation, while amygdala and globus pallidus are supported by anterior circulation. These results highlight how ASL measurements of blood perfusion can be used to infer additional features of the vasculature and inter-regional dependencies in blood perfusion in a data-driven manner.

Using these two maps we further construct a semantic space for how different vascular systems support cognition. We cross-correlate cortical maps from the Neurosynth meta-analytic

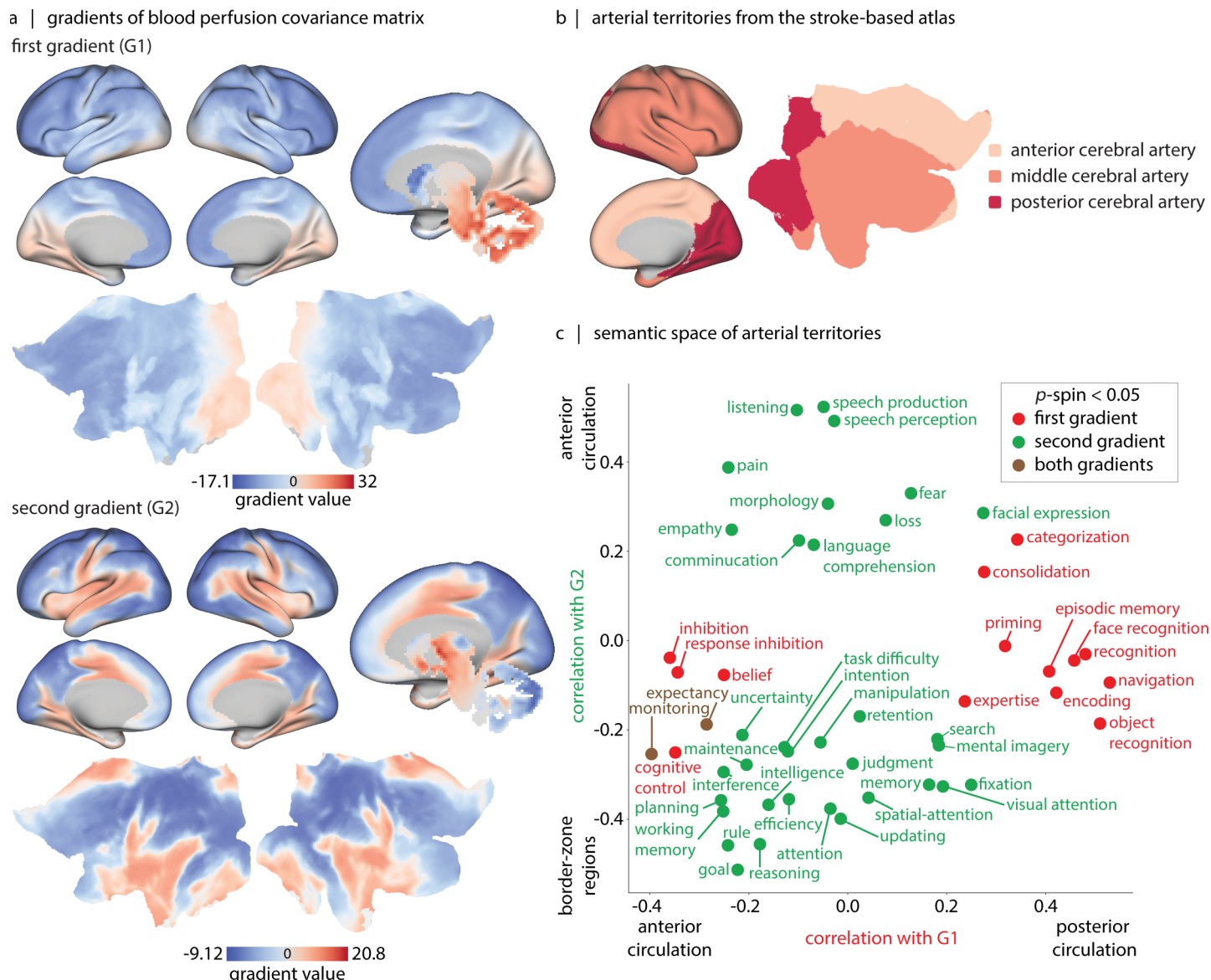

**Fig 7. Data-driven mapping and functional annotation of vascular territories.** (a) The first two gradients of blood perfusion covariance matrix are shown on the inflated and 2D flat cortical surfaces (fsLR), and subcortical regions are shown on a subcortical mask (from HCP-S1 200 release). (b) The arterial territories for posterior cerebral artery, middle cerebral artery and anterior cerebral artery from an independent stroke-based atlas [142,143]. (c) Correlation values between cortical gradient maps and Neurosynth meta-analytic atlas (x-axis: correlations with gradient 1; y-axis: correlations with gradient 2). Here, only functional/cognitive terms with correlation values exceeding thresholds determined by spatial auto-correlation-preserving permutation nulls ($p_{spin} > 0.05$) are annotated; for a list of 124 included Neurosynth functional/cognitive terms refer to S8 Table.

atlas with the two gradient maps (for a list of 124 included Neurosynth terms refer to S8 Table) [164–166]. The results suggest that posterior circulation (gradient 1) supports functions such as "navigation" [167], "face/object recognition" [168,169], "categorization" [170,171] and "episodic memory" [172,173]. Conversely, the anterior circulation (gradient 2) supports functions such as "speech", "language", "communication", "empathy", "fear" and "pain" [174] (Fig 7C). In other words, coordinated patterns of cerebral perfusion, reflecting vascular territories, are associated with distinct patterns of regional

specialization; consequently, this semantic space serves as a crude "lookup table" to match vascular territories to cognitive function.

## Discussion

In the present report, we analyze arterial spin labeling (ASL) data from 1305 (5–22 and 36–100 years) participants sourced from the HCP Lifespan studies, and investigate how blood perfusion fits within the multi-scale organization of the brain. At the molecular and cellular level, perfusion co-localizes with distinct cell types in cortical layer IV, metabolically demanding biological pathways, and receptors involved in vasomodulation. At the regional level, perfusion is greatest in regions with high metabolic demand and network hubs. Looking across the human lifespan, we find that cerebral perfusion undergoes considerable change, correlating with both micro-architectural changes and with individual differences in multiple health indicators. We find that cortical atrophy in multiple neurodegenerative diseases is most pronounced in regions with lower perfusion, highlighting the utility of perfusion topography as an indicator of transdiagnostic vulnerability. Finally, we show that ASL-derived maps can be used to identify major arterial territories, including those involved in the anterior and posterior circulation, providing new insights into how the vascular system supports human brain function.

Throughout the human body, blood supply is proportional to metabolic demands [175], partly as a result of common transcriptomic cues in vascular and cellular wiring [41,50,54,55, 176]. We find that this principle holds true in the brain as well [177]. Namely, regions with the greatest glucose and oxygen consumption at rest also tend to have the greatest perfusion. High-perfusion regions also exhibit greater connectivity within the resting-state functional connectome, suggesting that perfusion potentially reflects metabolic costs associated with inter-regional communication [91]. Notably, individuals with greater perfusion also display greater global functional connectivity, underscoring the importance of jointly examining these two features [178–180].

Blood perfusion is highly regionally heterogeneous. We find elevated perfusion in early sensory regions such as the early visual V1 strip, 3a and 3b somatosensory areas, and auditory areas [86], potentially reflecting metabolic costs associated with encoding sensory input [181]. We also observe elevated perfusion in a compact set of transmodal areas, including dorsolateral frontal cortex and highly-myelinated POS2. Interestingly, distinct clusters of high perfusion emerge on the map which are clearly distinct from neighboring areas, suggesting sharp transitions in vascular density (see Fig 2A) [98]. These clusters are located in heavily myelinated medial superior temporal (MST), middle temporal (MT), and lateral intraparietal (LIPv) areas [86], consistent with the notion that these heavily myelinated zones have greater cytochrome oxidase enzymatic activity [98,182,183].

At a finer spatial resolution, we also find that spatial variation in perfusion mirrors spatial variation in cellular organization. We find a close correspondence between layer IV density [88,92] and perfusion, consistent with the idea that layer IV is the most vascularized layer in mammalian cortex [93–103]. The link between perfusion and layer IV dovetails with the fact that layer IV is prominent in sensory cortex and in the lower half of the mid-dorsolateral prefrontal cortex (e.g., area 9–46) [184]. Furthermore, gene enrichment analyses revealed that the cerebral blood perfusion co-localizes with transcriptomic signatures of blood-brain barrier components, namely endothelial cells and pericytes which are involved in blood flow regulation [185–187]. The map also overlaps with In6a and In6b PVALB$^+$ cells, which are inhibitory interneurons characterized by high-frequency spiking patterns [23]. PVALB$^+$ cells exhibit greater mitochondrial volume compared to other cells in the brain [98,188], and are

correlated with vascular density in mouse models [103]. Through gene enrichment analysis, we also find that the cerebral blood perfusion pattern is linked with "mitochondrial organization", "responding to glucose stimulus", and "ion trans-membrane transport" (e.g., sodium and potassium). Notably, the link to ion transmembrane transport underscores the energy-intensive nature of maintaining the resting membrane potential in cells [189]. In sum, these results highlight the close link between the cellular environment and structural specialization of the brain and blood perfusion.

We further show that cerebral blood perfusion co-localizes with receptor densities of specific molecular signaling pathways [18,106,190]. Namely, the perfusion map aligns with vasoactive intestinal peptide (VIP) and endothelin (EDN) neuropeptide receptors, both of which are involved in regulating vascular diameter [191–197]. Additionally, we find correspondence with the $GABA_A$ neurotransmitter receptor, which mediates fast synaptic inhibition [198] and can dilate microvessels as a result of increased energy demands. Finally, we identify a correspondence between the $5HT_{1B}$ neurotransmitter receptor and blood perfusion, consistent with the fact that these receptors are directly expressed on arteries and regulate vasoconstriction [114–118]. Collectively, these findings suggest a coupling between molecular signaling pathways and the vasculature to modulate microvascular perfusion.

In addition to the cartography of cerebral blood perfusion, we investigate its dynamics across the lifespan. Cerebral blood perfusion increases during infancy and early years after birth potentially in response to heightened synaptogenesis and myelination, and reaches a maximum at 4–8 years of age [127–133]. Within the HCP-Development age range (5–22 years), we capture the decline phase for blood perfusion following this extremum, which may be driven by synaptic pruning [134]. Indeed, there is a positive association between the age-related decline in cerebral blood perfusion and age-related decline in cortical thickness. Notably, decrease in blood perfusion is more pronounced in transmodal cortex, likely due to the greatest overproduction and slowest elimination of dendritic spines in these areas [135,136]. Additionally, within this age window, well-known sex differences in cerebral blood perfusion—where females typically exhibit higher cerebral blood perfusion than males—begin to emerge, possibly influenced by sex hormones [199,200].

Incorporating the HCP-Aging dataset, we also examine changes in cerebral blood perfusion during aging (36–100 years). Atherosclerosis, impaired cardiac output, and elimination of brain vasculature all affect cerebral blood perfusion and contribute to functional impairments in aging [190,201–207]. We find greater age-related reduction in perfusion in border areas between major arterial territories, in line with the fact that these areas have reduced vascular density and smaller vessels and are vulnerable to embolism [140,154,156,208]. Going further, we also show that additional physiological factors are associated with age-related reductions in perfusion, including mean arterial pressure (MAP), BMI, glucose, and HbA1c, reminiscent of the fact that metabolic syndrome exacerbates blood perfusion reduction, above and beyond the effect of sex and age [209,210]. Indeed, high glucose (as well as obesity and insulin resistance) can lead to microangiopathy, atherosclerosis [211,212] and endothelial dysfunction through activation of protein kinase C, overexpression of growth factors and/or cytokines, and oxidative stress [213]. Meanwhile, our results point to the possible atheroprotective effect of HDL; HDL is involved in suppressing vascular-LDL accumulation, inflammation, and endothelial damage [214,215]. The tight coupling between plasma biomarkers and cerebral blood perfusion underpins the importance of considering body-brain interactions in the aging population.

To evaluate whether low baseline blood perfusion also confers vulnerability to pathology in symptomatic aging, we examined the correspondence between disease atrophy patterns and the regional differences in blood perfusion. Regions of low blood perfusion co-localize

with atrophy patterns in late-onset Alzheimer's disease (LOAD), dementia with Lewy bodies (DLB) and TDP-43C pathologies. Vascular impairments and disturbances in blood flow delivery are noted in LOAD patients prior to symptom onset [47,216]. Interestingly, we did not observe an association between cerebral blood perfusion and early-onset Alzheimer's disease (EOAD) atrophy pattern, perhaps because EOAD is less associated with vascular risk factors compared to LOAD [217–219]. Finally, TDP-43 build-up has been documented as a stress response to hypo-perfusion in mice models [220,221], potentially explaining the link between TDP-43C-related atrophy and low perfusion. These results suggest that regions with suboptimal perfusion are rendered more vulnerable to the build-up of neurotoxins and the onset of atrophy, contributing to a growing appreciation for the importance of clearance of neurotoxins in multiple neurodegenerative diseases [216,222]. Although these associations are derived from normative perfusion, they point to a pervasive influence of perfusion on healthy and non-healthy aging that should be studied more directly using perfusion measurements in individual patients.

Finally, the present report demonstrates how inter-individual differences in cerebral blood perfusion can be used to investigate inter-regional coordination of blood flow across the brain and to derive additional methodologically useful features. In particular, we show that it is possible to delineate arterial territories with high spatial resolution, including posterior circulation (basilar artery and posterior cerebral artery), anterior circulation (anterior and middle cerebral arteries) and border-zones. These findings build on multiple empirical reports on the construction of digital arterial atlases. The conventional approach is to construct arterial territories based on stroke outcomes [143,223–227], which is potentially biased by stroke probability and stroke cohort [228–232]. Alternative methods are either invasive, such as intracarotid amobarbital single-photon emission computerized tomography (SPECT) imaging [233], or involve targeted selection of specific arteries, such as selective arterial spin labeling—which additionally introduces the possibility of signal contamination by other arteries [234–236]. The present report uses data from typical control participants and a non-invasive data-driven approach to project high-dimensional ASL data to a low-dimensional space, and is conceptually similar to a previous report applying independent component analysis to ASL ($N$ = 92) [163]. Furthermore, we demonstrate how the resulting arterial "gradients" can be functionally annotated using an independent meta-analytic atlas. The resulting semantic space opens the possibility to more directly link vascular impairments with physiological processes.

The present findings should be considered with respect to several methodological limitations. First, ASL imaging has a low signal-to-noise ratio, which can reduce data robustness at the individual-participant level. To help address this issue, we apply principal component analysis and use the first principal component for the cartography of cerebral blood perfusion. Second, perfusion is contextualized with respect to canonical features of brain structure and function measured in different participants, such as microarray gene expression, PET neurotransmitter receptors, MRI developmental surface expansion and Neurosynth activation maps. This methodological limitation highlights the need for more extensive multi-modal imaging and biological assays in healthy participant datasets [237]. Third, when estimating the association of perfusion with other biological features, age-related changes of perfusion in development and aging, and relating vascular territories to functional annotations, we focused on cortical regions. However, we provide quantification of subcortical perfusion values, age-related effects and perfusion covariance gradient values, and make the whole-brain maps openly available for further subcortex-focused analysis. Lastly, it is noteworthy that a complementary avenue to study brain physical vascular system is by using imaging methods such as Time-of-flight Magnetic Resonance Angiography (TOF-MRA) [238–240]. As the

imaging techniques and segmenting tools advance, we envision that studies will integrate vascular network and perfusion measurements to investigate brain vascular organization.

## Materials and methods

All codes used to perform the analyses are available on GitHub at https://github.com/netneurolab/Farahani_Blood_Perfusion/ and on Zenodo at https://zenodo.org/records/15708107 (DOI: 10.5281/zenodo.15708107). The group-based score maps and the arterial territory maps are provided in the same directory.

### Dataset: Demographics

Data from a total of 1305 participants in the HCP Lifespan studies (2.0 Release) are analyzed in this study [73–75]. Specifically, data from 627 participants (337 females) from HCP-D (5–22 years) and 678 participants (381 females) from HCP-A (36–100 years) are included. All study procedures are conducted in accordance with the principles expressed in the Declaration of Helsinki and are approved by the Institutional Review Board at Washington University in St. Louis (HCP-D IRB ID#: 201603135; HCP-A IRB ID#: 201603117).

### Dataset: Image acquisition

All HCP Lifespan imaging data is acquired using a 3.0 Tesla Prisma scanner (Siemens; Erlangen, Germany) and a 32–channel head coil.

T1-weighted image is acquired using a multi-echo magnetization-prepared rapid gradient echo (MPRAGE) sequence with the following parameters: repetition time (TR) = 2 500 ms, inversion time (TI) = 1 000 ms, echo times (TE) = 1.8/3.6/5.4/7.2 ms, spatial resolution = $0.8 \times 0.8 \times 0.8$ mm$^3$, number of echoes = 4 and flip angle = 8°. T2-weighted image is acquired using a 3D sampling perfection with application-optimized contrasts using different flip angle evolutions (SPACE) sequence, with the same spatial resolution as the T1-weighted image. The T2-weighted image acquisition parameters are: TR = 3 200 ms, TE = 564 ms and turbo factor = 314 ms. Both T1- and T2-weighted scans cover a sagittal field of view of $256 \times 240 \times 166$ mm, with a matrix size of $320 \times 300 \times 208$ slices. Slice oversampling of 7.7% is used, as is 2-fold in-plane acceleration (GRAPPA) in the phase encode direction and a pixel bandwidth of 744 Hx/Px. T1- and T2-weighted data are used for registration purposes within the arterial spin labeling and functional MRI preprocessing pipelines [73].

The blood perfusion data comes from arterial spin labeling (ASL) magnetic resonance imaging (MRI). In this imaging technique, the magnetization of arterial blood in the neck is flipped (or "labeled") using a train of short block radiofrequency pulses. Images of the brain containing the magnetic "label" are then acquired following a delay that allows the labeled blood to reach and perfuse brain tissue (i.e., post-labeling delay; PLD). In addition, a "control" image with no magnetization inversion is also acquired. The difference between the "label" images and the "control" image is proportional to cerebral blood perfusion and is not affected by the static spins. Through this imaging, arterial transit time (ATT) which represents the time it takes for labeled blood to reach brain tissue is also estimated [85,241]. In HCP Lifespan studies, ASL data acquisition is based on a pseudo-continuous arterial spin labeling (pCASL) and a 2D multiband (MB)-echo-planar imaging (EPI) sequence. Pseudo-continuous ASL data are acquired with labeling duration of 1 500 ms and five post-labeling delays of 200 ms, 700 ms, 1 200 ms, 1 700 ms, and 2 200 ms, containing 6, 6, 6, 10, and 15 control-label image pairs, respectively. To calibrate perfusion measurements into units of ml/100g/min, two PD-weighted M0 calibration images (TR > 8 $s$) are acquired at the end of

the pCASL scan. Other sequence parameters include: spatial resolution = 2.5 × 2.5 × 2.5 mm³, and TR/TE = 3 580/18.7 ms. For susceptibility distortion correction, two phase-encoding-reversed spin-echo images are also acquired. Participants 8 years or older, view a small white fixation crosshair on a black background, and the 5–7-year-old participants view a movie during the scan time (5.5 min). The ASL preprocessing steps are described in the following section.

For participants in the HCP lifespan studies, resting-state functional data with blood-oxygen-level-dependent (BOLD) contrast is also acquired. The sequence used to acquire the functional data is a 2D multi-band gradient-recalled echo (GRE) echo-planar imaging (EPI). The scan parameters include: TR/TE = 800/37 ms, flip angle = 52°, spatial resolution = 2.0 × 2.0 × 2.0 mm³. For participants 8 years and older, functional scans are acquired in pairs of two runs (four runs in total per participant, each run lasting 6.5 min), with opposite phase encoding polarity so that the functional MRI data in aggregate is not biased toward a particular phase encoding polarity (two runs have the phase encoding of anterior-to-posterior (AP) and two runs have the phase encoding of posterior-to-anterior (PA)). For participants between 5–7 years old, six scans with a shorter duration of 2.5 min are acquired. During resting-state functional MRI scanning, participants view a small white fixation crosshair on a black background. The functional MRI minimal preprocessing steps that have been applied to the data are detailed in [242].

## Dataset: ASL data preprocessing

The ASL data preprocessing is conducted following the HCP ASL Pipeline available at https://github.com/physimals/hcp-asl/ [241]. To run this pipeline we use QuNex platform (singularity container, version 0.99.1) [243].

Briefly, raw ASL and calibration images are first registered to participant anatomical space. This registration transform is merged with motion correction, susceptibility distortion, and gradient distortion transforms and applied simultaneously to the raw ASL and calibration images using the `regtricks` library. This approach minimizes repeated interpolations and thus reduces partial volume effects. Next, intensity bias and banding corrections are applied to the structurally-aligned ASL data. In the presence of head motion, voxels traveling between neighboring slices during the acquisition receive differing intensity scaling during the banding corrections, which can lead to spurious signal after label-control subtraction. To address this effect, general linear model (GLM)-based approach for motion-aware subtraction of banded and background-suppressed ASL data, previously introduced by Suzuki et al. [244], is used. The subtracted time-series data is then used for perfusion estimation. Cerebral blood perfusion and arterial transit time (ATT) are estimated using a variational Bayesian method via the `oxford_asl` script [245]. The `aslrest` Buxton model with cerebral blood perfusion, ATT and macrovascular components is used [246,247]. A normal distribution prior with a mean of 1 300 ms is imposed on ATT and an automatic relevancy determination prior is used on macrovascular perfusion to remove this component from non-arterial voxels. Slice-timing correction is performed by adjusting post-labeling delays in each voxel according to its slice timing offset, and perfusion is converted from arbitrary units into ml/100g/min using the mean signal value of cerebral blood flow in the lateral ventricles from the calibration image [248]. Lastly, partial volume effect (PVE) is corrected using a spatial variational Bayes method implemented in `oxford_asl` [247]; the required partial volume estimates are obtained using Toblerone operating on the FreeSurfer-derived cortical surfaces and subcortical segmentations [249,250]. For PVE correction, a normal distribution prior with a mean of 1 300 ms for grey matter ATT and a mean of 1 600 ms for white matter ATT is used.

Volumetric cerebral blood perfusion and ATT maps from `oxford_asl` are produced in both ASL-gridded T1-weighted space and MNI152–2 mm template space, via the FNIRT-based registration produced by the HCP structural processing pipeline. To generate output maps in grayordinates space (CIFTI format), the volumetric outputs of `oxford_asl` are projected onto the individual's native cortical surface using the HCP ribbon-constrained method, registered with MSMAll multi-modal areal-feature-based registration, resampled to a common surface mesh, and finally smoothed with a 2 mm full-width half-maximum kernel using a surface-constrained method [86,242,251,252]. The surface and MNI outputs (masked to consider subcortical structures only) are combined to produce the final CIFTI grayordinates files, which serve as the input data format for subsequent analyses in this study.

## Dataset: Biomarkers

In addition to imaging data, the HCP-A dataset includes a comprehensive array of participant-specific blood laboratory results (plasma measurements) and bodily vitals [75]. The plasma panel includes common health indicators such as total protein, glucose, insulin, hemoglobin A1c (HbA1c), triglycerides, low-density lipoprotein (LDL), high-density lipoprotein (HDL), total cholesterol, albumin, bilirubin, creatinine, urea, chloride, sodium, potassium, calcium, vitamin D, and $CO_2$ content, as well as liver metabolic enzymes including alanine aminotransferase (ALT), aspartate aminotransferase (AST), and alkaline phosphatase (ALP). Multiple hormonal measures are also obtained including serum estradiol, testosterone, Luteinizing hormone (LH), and follicle-stimulating hormone (FSH). This extensive biochemical data is available for 597 out of 678 participants in the HCP-A dataset who have corresponding ASL data. Furthermore, participants' height and weight data are provided, enabling the calculation of body mass index (BMI). Systolic and diastolic blood pressure/pulse values are also recorded, from which the mean arterial pressure (MAP) is computed. Throughout the manuscript, we refer to the combination of plasma measurements, vitals (including BMI and MAP) and age as "biomarkers". S20 Fig presents scatter-plots illustrating the relationship between participants' raw measures and age.

Among the 597 participants (329 females) with available biomarker data, 2 have missing values for BMI, 6 for MAP, 4 for HbA1c, and 3 for hormonal measures. In these instances, the missing values are imputed using the mode of the respective measure within each biological sex group to prevent not-a-number values in the PLS analysis.

## Atlases

For all parcel-wise cortical analysis, we apply the Schaefer-400 functional atlas [126].

For quantifying cortical perfusion scores (S2 Table), we use the multi-modal Glasser parcellation; this parcellation divides each hemisphere into 180 distinct parcels based on function, cortical microstructure, and connectivity [86]. We used this parcellation only when showing areal borders in Fig 2A. This is because the Glasser atlas integrates multi-modal information and offers a more anatomically precise representation of cortical architectonic boundaries than the Schaefer atlas, which focuses solely on resting-state functional signal homogeneity [126]. S2 Table and Fig 2A are the only instances where multi-modal Glasser parcellation is used.

For quantifying subcortical perfusion scores (S7 Fig), age-effect (S11 Fig), and perfusion covariance gradients (S23 Fig), we use the Tian functional atlas (S4 version; including 54 parcels; see S3 Table) [87].

### Functional connectivity

Functional connectivity quantifies the synchronization of fluctuations in BOLD signals across brain regions. Functional MRI data is preprocessed using the HCP Minimal Preprocessing Pipeline. For detailed preprocessing steps, refer to the cited reference [242]. The vertex-wise functional MRI time-series are initially demeaned and subsequently parcellated using the Schaefer-400 atlas [126]. The parcellated time-series are then $z$-scored and concatenated across participant runs. Each participant's unified time series is used to derive functional connectivity matrix. The functional connectome is computed by calculating the Pearson correlation coefficient between pairs of regional time-series.

To construct a representative map of functional connectivity strength, first, we compute the absolute value of all functional connectivity edges and sum the edges connected to each region for each participant. We then $z$-score the region-wise strength vector per participant, and combine data of all participants into a single matrix. Principal component analysis (PCA) is applied on the obtained matrix. Finally, the first score map (explaining 52.7% of the variance) is compared with the normative map of blood perfusion in Fig 2C.

### Cortical thickness

Cortical thickness quantifies the width of cortical grey matter. The individual-level cortical thickness maps are derived through the HCP Minimal Preprocessing Pipeline [242]. In short, this pipeline uses both participant-specific bias-corrected T1-/T2-weighted structural data to segment the cortical grey and white matter and perform a surface reconstruction using FreeSurfer. Next, cortical thickness is estimated as the geometric distance between the white and pial surfaces.

### Cortical layer IV

Cortical layer IV pattern is estimated as the average expression map of 5 genes preferentially expressed in human granular layer IV [92]. Gene maps for "COL6A1", "CUX2", "TRMT9B", "GRIK4", and "RORB" are obtained from abagen toolbox, publicly available at https://github.com/rmarkello/abagen/ [253]. We incorporated the gene maps parcellated based on Schaefer-400 atlas throughout the manuscript. Following Burt et al. [88], each gene map is standardized using $z$-scoring, and their average is calculated to obtain the layer-specific cortical pattern.

### Neurotransmitter receptors

We use PET-derived receptor density data for 7 neurotransmitter receptors that have previously been associated with vasomodulation [18]. These receptors cover three neurotransmitter systems, including serotonin (5-HT$_{1A}$ [19], 5-HT$_{1B}$ [254–260], 5-HT$_{2A}$ [19], 5-HT$_4$ [19], 5-HT$_6$ [261,262]), glutamate (NMDA) [263–265], and GABA (GABA$_A$) [20]. The receptor maps are originally collected by Hansen et al. [18] and are downloaded from the neuromaps toolbox [113] (available at https://github.com/netneurolab/neuromaps/). Each of the volumetric PET tracer images is parcellated based on the Schaefer-400 atlas [126]. Methodological details about each tracer can be found in S6 Table.

### Neuropeptide receptors

Neuropeptide receptor maps are estimated using bulk tissue microarray expression data collected from six post-mortem brains (1 female; age: 24–57, mean age: 42.50±13.38) [106]. This data is provided by the Allen Human Brain Atlas (AHBA) (https://human.brain-map.

org/) [13] and is processed using the abagen toolbox, publicly available at https://github.com/rmarkello/abagen/ [253], yielding a map for each gene in the parcellated MNI template (Schaefer-400 [126]). All four included receptor genes have high differential stability across donors (threshold of 0.1). Note that only two donors from the AHBA have tissue samples taken from the right hemisphere. This irregular sampling results in limited spatial coverage of expression in the right hemisphere; to resolve this, tissue samples are mirrored bilaterally across the left and right hemispheres. Consequently, the final gene expression profile for each region is estimated as the mean of both ipsilateral samples (from all six donors with left hemisphere samples) and contralateral samples (from the two donors with right hemisphere samples) [266].

## Metabolic maps

We use the neuromaps toolbox (available at https://github.com/netneurolab/neuromaps/) to obtain PET-based energy metabolism maps [113]. The maps are initially provided in the fsLR (4k) surface space; we use `fslr_to_fslr` function to transform them into the fsLR (32k) surface space. We next parcellate the maps using the Schaefer-400 functional atlas [126].

Cerebral metabolic rates for glucose ($CMR_{Glc}$), oxygen ($CMR_{O2}$), and cerebral blood flow are measured in 33 normal right-handed adults. The data is acquired when participants are in the resting awake state with eyes closed. The cohort is composed of 14 males and 19 females in the age range of 20–33 years old (mean 25.4 ± 2.6 years). Regional $CMR_{Glc}$ is measured using the administration of [$^{18}$F]-labeled fluorodeoxyglucose. Regional $CMR_{O2}$ is measured by inhalation of [$^{15}$O]-labeled oxygen. Regional cerebral blood flow is measured using the administration of [$^{15}$O]-labeled water, as detailed in [30].

## Developmental expansion map

The developmental surface expansion map is obtained from the neuromaps toolbox [113] (available at https://github.com/netneurolab/neuromaps/). The data is initially provided in the fsLR (164k) surface space; we employ the `fslr_to_fslr` function to transform it into the fsLR (32k) surface space. We then parcellate the resulting map using the Schaefer-400 functional atlas [126].

The map is generated by comparing the cortical surface reconstructions from 12 healthy term-born human infants (6 males and 6 females, mean gestational age = 39 weeks) and corresponding reconstructions from 12 healthy young adults (6 males and 6 females, 18–24 years). Further details on the methodology can be found in [125].

## Neurosynth meta-analytic term maps

Probabilistic measures of the association between voxels and cognitive processes are obtained from Neurosynth [164], a meta-analytic tool that synthesizes results from more than 14,000 published neuroimaging studies by searching for high-frequency keywords that are published alongside their associated voxel coordinates (https://github.com/neurosynth/neurosynth/, using the volumetric association test maps [166]). This measure of association is the probability that a given cognitive process is reported in the study if there is activation observed at a given voxel. Although 1 334 terms are reported in the Neurosynth engine, we focus our analysis primarily on cognitive function and, therefore, limit the terms of interest to cognitive and behavioral terms. To avoid selection bias, the terms are selected from the Cognitive Atlas, a public ontology of cognitive science [165], which includes a comprehensive list of neurocognitive terms. We end up incorporating 124 terms, ranging from umbrella terms ("attention"

and "emotion") to specific cognitive processes ("visual attention" and "episodic memory"), behaviors ("eating" and "sleep") and emotional states ("fear" and "anxiety"). The coordinates reported by Neurosynth are parcellated according to the Schaefer-400 atlas [126]. The full list of Neurosynth functional/cognitive included terms is listed in S8 Table.

## Null models

To assess the effect of spatial auto-correlation on spatial associations between two cortical brain maps, we use the so-called spatial auto-correlation preserving permutation tests, commonly referred to as "spin tests" [267]. Briefly, brain phenotypes are projected to spherical projection of the fsaverage surface. This involves selecting the coordinates of the vertex closest to the center of mass for each parcel. These parcel coordinates are then randomly rotated, and original parcels are reassigned to the value of the closest rotated parcel ($N$ repetitions; throughout the manuscript $N = 1\,000$, unless stated otherwise). For parcels where the medial wall is the closest, we assign the value of the next closest parcel instead. Following these steps, we obtain a series of randomized brain maps that have the same values and spatial auto-correlation as the original map, while the relationship between values and their spatial location has been permuted. These maps are then used to generate null distributions of desired statistics. Throughout the manuscript, whenever the spatial correspondence between two brain maps is tested for, "spin test" is carried out for maps parcellated with the Schaefer-400 cortical atlas [126]. Notably, this parcellation is chosen because it divides the cortex into relatively homogeneous parcel sizes, and its number of parcels is comparable to the estimated number of distinct human neocortical areas [268,269].

## Gene category enrichment

Biological pathways that are correlated with the blood perfusion score map are identified using a gene category enrichment analysis (GCEA). Cortical maps for biological pathways are defined according to the gene expression data coming from the AHBA [13]. Transcriptomic data is preprocessed and mapped to parcellated brain regions (Schaefer-400 [126]) using the abagen toolbox, publicly available at https://github.com/rmarkello/abagen/ [253]. To perform the enrichment analysis, we use the ABAnnotate Matlab-based toolbox, publicly available at https://github.com/LeonDLotter/ABAnnotate/ [270]. The package is adapted from the toolbox initially developed by Fulcher et al. [105] (https://github.com/benfulcher/GeneCategoryEnrichmentAnalysis/). The GCEA procedure assesses whether genes in a particular category are more correlated with a given brain phenotype than a random phenotype with comparable spatial auto-correlation (ensemble-based null model) [105].

To address spatial auto-correlation effects, 50 000 spatially auto-correlated null maps are generated from the blood perfusion score map using the neuromaps toolbox (method = "vasa" [113,271]) and are inputted to ABAnnotate package for the testing procedure. After matching category and Allen Human Brain genes based on gene symbols and removing the genes with differential stability lower than 0.1, the Pearson correlations between the blood perfusion score map, the null maps, and all gene expression maps are calculated. For each null map and each category, null category scores are obtained as the mean $z$-transformed correlation coefficients. Positive-sided $p$-values are determined by comparing the "true" category scores with the null distribution, with subsequent False Discovery Rate (FDR) correction applied. For gene-category annotations, we use the GO biological processes (with more than 30 annotated genes) [104] as well as the cell-type categories introduced by Lake et al. [23].

Across the 401 included biological processes, 88 remained statistically significant following multiple comparisons correction (FDR) (for full list refer to S4 Table). We sort the significant terms based on the corresponding category scores and show the top 10 processes in Fig 3A.

## Generalized additive model for location, scale and shape

Generalized additive model for location, scale and shape (GAMLSS) is implemented using the gamlss R package available at https://github.com/gamlss-dev/ [272]. We use a sex-stratified GAMLSS modeling approach to characterize nonlinear normative cerebral blood perfusion trajectories in development and aging. In this framework, cerebral perfusion in a given parcel or across the whole brain is assumed as a random variable ($Y$) following the generalized gamma (GG) distribution:

$$Y \sim GG(\mu, \sigma, \nu) \tag{1}$$

here, GG distribution parameters—location ($\mu$), scale ($\sigma$), and shape ($\nu$)—can be modeled as functions of explanatory variable (age). In our analysis, $\mu$ and $\sigma$ are modeled as the fractional polynomial functions of the explanatory variable, and $\nu$ is set as a constant. Specifically, we used the fp() function to determine the best-fitting two-term fractional polynomials from the predefined set of power values: –2,–1,–0.5,0,0.5,1,2,3. The optimal set of powers is chosen automatically through iterative model fitting to best capture nonlinear relationships between age and distribution parameters.

## Partial least squares

The partial least squares (PLS) component analysis is performed using the behavioral_pls function implemented in the pyls toolbox available at https://github.com/netneurolab/pypyls/. The objective of PLS analysis is to identify the relationship between two data matrices [144,145]. In this study, the two matrices represent blood perfusion maps (participants×regions) and biomarkers (participants×measures). Both matrices are first $z$-scored by subtracting the mean from each column (feature) and dividing by the standard deviation. The covariance (correlation) between normalized cerebral blood perfusion (**X**) and biomarkers (**Y**) is then computed. The resulting covariance matrix is subjected to singular value decomposition (SVD):

$$\mathbf{X'Y} = \mathbf{USV'} \tag{2}$$

where **U** and **V** are orthonormal matrices of left and right singular vectors and **S** is a diagonal matrix comprising singular values. Each triplet of a left singular vector, a right singular vector, and a singular value constitutes a latent variable. Singular vectors weight the contribution of original features (regional blood perfusion values and biomarkers) to the overall multivariate pattern. To assess the extent to which individual participants express these blood perfusion or biomarker patterns (the multivariate pattern captured by the latent variable), participant-specific brain and biomarker scores are calculated. Scores are computed by projecting the original data onto the respective singular vector weights, such that each individual is assigned a brain score and a biomarker score:

$$\text{cerebral blood perfusion score} = \mathbf{XU} \tag{3}$$

$$\text{biomarker score} = \mathbf{YV} \tag{4}$$

here, the scores indicate the degree to which a participant expresses each brain pattern and biomarkers pattern. Finally, loadings are computed by calculating the correlations between

participant-specific biomarker (or brain measure) and the respective score pattern (loadings are shown in Fig 5D and 5E).

The proportion of covariance accounted for by each latent variable is quantified as the ratio of the squared singular value to the sum of all squared singular values. The statistical significance of each latent variable is determined by permutation testing. This involves randomly permuting the order of observations (i.e., rows) of data matrix $\mathbf{X}$ for a total of 1 000 repetitions, followed by constructing a set of "null" brain-biomarker correlation matrices. These "null" correlation matrices are then subjected to SVD, to generate a distribution of singular values under the null hypothesis that there is no association between brain perfusion pattern and participants' biomarkers. A non-parametric $p$-value can be estimated for a given latent variable as the probability that a permuted singular value exceeds the original, non-permuted singular value.

The reliability of individual biomarker contribution to the model is evaluated using bootstrap resampling. Participants (rows of data matrices $\mathbf{X}$ and $\mathbf{Y}$) are randomly sampled with replacement across 1 000 repetitions, resulting in a new set of correlation matrices that are subsequently subjected to SVD. This procedure generates a sampling distribution for each individual weight in the singular vectors. For each biomarker, a bootstrap ratio is computed as the ratio of its singular vector weight to its bootstrap-estimated standard error. High bootstrap ratios indicate biomarkers that significantly contribute to the latent variable and are stable across participants.

Finally, we use cross-validation to assess the generalizability of PLS results [273,274]. Specifically, we assess the out-of-sample correlation between cortical blood perfusion and biomarker scores. We use 1 000 randomized train-test splits of the dataset; in each random split, we allocate 80% of the data for training and the remaining 20% for testing. In each iteration, PLS is applied to the training data ($\mathbf{X}_{train}$ and $\mathbf{Y}_{train}$) to estimate singular vector weights ($\mathbf{U}_{train}$ and $\mathbf{V}_{train}$); test data is then projected onto these derived weights to compute participant-specific scores ($\mathbf{X}_{test}U_{train}$ and $\mathbf{Y}_{test}V_{train}$). The correlation between brain and biomarker scores is evaluated for the test sample. Ultimately, this procedure leads to 1 000 correlation values and establishes a distribution of out-of-sample correlation values. To assess the statistical significance of these out-of-sample correlation values, we conduct permutation tests (1 000 repetitions). During each permutation, we shuffle the matrix rows and repeat the analysis to create a null distribution of correlation coefficients between cerebral blood flow and biomarker scores in the test sample. This null distribution is then used to estimate a non-parametric $p$-value, by calculating the proportion of null correlation coefficients that are greater than or equal to the mean original out-of-sample correlation coefficient. For the first latent variable, mean of out-of-sample correlations between cerebral blood perfusion and biomarker scores is equal to $r = 0.30$ for male ($p = 0.02$), and $r = 0.34$ ($p = 2.99 \times 10^{-3}$) for female groups.

## Neurodegenerative disease atrophy maps

Data for eight neurodegenerative diseases are provided by Harper et al. [159], and are downloaded from https://neurovault.org/collections/ADHMHOPN/. T1-weighted MR images are collected from 186 individuals with a clinical diagnosis of dementia and histopathological (postmortem or biopsy) confirmation of underlying pathology, along with 73 healthy controls. Data is averaged across participants per condition: 107 had a primary Alzheimer's disease (AD) diagnosis—68 early-onset [<65 years at disease onset], 29 late-onset [≥65 years at disease onset], 10 presenilin 1 (PS1) mutation carriers; 25 with dementia with Lewy bodies (DLB), 11 with three-repeat-tauopathy (3Rtau), 17 with four-repeat-tauopathy (4Rtau), 12

FTLD-TDP type A (TDP-43A), and 14 FTLD-TDP type C (TDP-43C). Imaging data is collected from multiple centers on scanners from three different manufacturers (Philips, GE, and Siemens) using a variety of different imaging protocols. Magnetic field strength varies between 1.0T ($N = 15$ scans), 1.5T ($N = 201$ scans), and 3T ($N = 43$ scans). Pathological examination of brain tissue is conducted according to the standard histopathological processes and criteria in use at the time of assessment at one of four centers: the Queen Square Brain Bank, London; Kings College Hospital, London; VU Medical Center, Amsterdam; and Institute for Ageing and Health, Newcastle.

Tissue volume loss is measured by voxel-based morphometry (VBM) applied to structural T1-weighted MR images, and expressed as a $t$-score per voxel. Atrophy maps are statistically adjusted for age, sex, total intracranial volume, MRI strength field and site. Ethical approval for this retrospective study is obtained from the National Research Ethics Service Committee London-Southeast [159].

## Gradient maps

We use the python version of the BrainSpace toolbox (https://github.com/MICA-MNI/BrainSpace/) to find the gradients of the blood perfusion covariance matrix [275].

Participant-specific perfusion maps are originally produced as grayordinates files (including 91 282 vertices/voxels). For computational efficiency, we first down-sample participants' data. For the cortical vertices, we resample the data from the original fsLR 32k space to fsLR 4k space, using the `fslr_to_fslr` function implemented in neuromaps; we further resample the grayordinates subcortical data to 4 mm resolution using the `resample_img` function in Nilearn python package. The down-sampling step reduces participant-specific data-points to 12 017 vertices/voxels. Subsequently, we $z$-score the data of individual participants and then $z$-score each region across individuals. Based on normalized down-sampled data, we compute the covariance matrix across participants ($12\,017 \times 12\,017$). A Diffusion mapping manifold algorithm (with default parameters) is used to derive the first five gradients of the covariance matrix, the *Lambda* values (eigenvalues) corresponding to these are 11.55, 7.05, 4.16, 2.15, and 2.06. The first two gradients are selected for further investigation.

## Datasets and code

The code to obtain the results within this manuscript mainly relies on open-source Python packages, including NumPy (version 1.21.6) [276,277], SciPy (version 1.7.3) [278], pandas (version 1.3.5) [279], seaborn (version 0.12.2) [280], Matplotlib (version 3.5.3) [281], statsmodels (version 0.13.5) [282], bctpy (version 0.6.1) [283], Nilearn (version 0.10.1, see [284]), NiBabel (version 4.0.2) [285], netneurotools (version 0.2.3) [286] and rpy2 (version 3.5.17) [287].

Furthermore, the neuromaps toolbox (version 0.0.4) [113], used to obtain neurotransmitter receptor maps, energy consumption maps and developmental surface expansion map, is available at https://netneurolab.github.io/neuromaps/. The abagen toolbox (version 0.1.3) [253] for processing the AHBA human transcriptomic dataset [13] is available at https://abagen.readthedocs.io/. The ABAnnotate Matlab-based toolbox (version 0.1.1), to perform the gene enrichment analysis [105], is available at https://github.com/LeonDLotter/ABAnnotate/. Parcellation atlases, including the Schaefer-400 [126] and Tian-S4 subcortical atlas [87], can be obtained from https://github.com/yetianmed/subcortex/tree/master/Group-Parcellation/3T/. Glasser multi-modal parcellation (version 1.0) [86] can be obtained from balsa website at https://balsa.wustl.edu/study/RVVG/ [288]. The GAMLSS models are implemented using GAMLSS R package (version 5.4-22) available at https://github.com/

gamlss-dev/ [272]. The PLS analysis is performed using the pyls toolbox (version 0.0.1) available at https://github.com/netneurolab/pypyls/. Disease atrophy maps [159] are available at https://neurovault.org/collections/ADHMHOPN/. Gradient analysis of blood perfusion covariance matrix is performed using the BrainSpace toolbox (version 0.1.10) [275] available at https://github.com/MICA-MNI/BrainSpace/. All brain plots in the manuscript are visualized using Connectome Workbench (version 1.5.0), available at https://www. humanconnectome.org/software/get-connectome-workbench/ [289].

## Acknowledgments

We thank Vincent Bazinet, Filip Milisav, Yigu Zhou, Tahmineh Taheri, and Moohebat Pourmajidian for their comments and suggestions on the manuscript. We thank all investigators and staff members of the Human Connectome Project consortium for their invaluable efforts in data collection.

## Supporting information

**S1 Fig. Mapping white matter blood perfusion across participants.** Each dot corresponds to a participant's mean blood perfusion level within the white matter mask defined as part of the HCP ASL preprocessing pipeline (male: blue, female: red). Top: Sex-stratified generalized additive models for location, scale and shape (GAMLSS) are used to model age-related changes in blood perfusion across the human lifespan. Bottom: Statistical comparisons of white matter blood perfusion between males and females are conducted within each age bin (13–15 years: $t = -3.76$, $p = 2.62 \times 10^{-4}$; 15–17 years: $t = -4.97$, $p = 3.43 \times 10^{-6}$; 19–21 years: $t = -4.73$, $p = 1.36 \times 10^{-5}$; 21–23 years: $t = -2.96$, $p = 5.29 \times 10^{-3}$; 35–47 years: $t = -7.26$, $p = 1.75 \times 10^{-11}$; 47–57 years: $t = -6.69$, $p = 9.47 \times 10^{-11}$; 57–67 years: $t = -3.86$, $p = 1.88 \times 10^{-4}$; 67–77 years: $t = -3.39$, $p = 9.98 \times 10^{-4}$; >77 years: $t = -4.61$, $p = 1.01 \times 10^{-5}$; other age bins: $p > 0.05$).
(PDF)

**S2 Fig. Sex-difference in cerebral blood perfusion.** (a) The vertex/voxel-wise mean blood perfusion maps across all participants (HCP-D and HCP-A), stratified by biological sex. Maps are shown on the inflated and 2D flat cortical surfaces (fsLR) and on a T2-weighted group-average template [290]. (b) Left: Perfusion is higher in female brain compared to the male brain ($t = 9.27$, $p_{\text{two-sided}} = 7.21 \times 10^{-20}$). Each dot corresponds to a participant's whole brain blood perfusion (male: blue, female: red). Right: Correlation between female ($x$-axis) and male mean blood perfusion ($y$-axis) ($r = 0.99$). Each dot corresponds to mean blood perfusion for a vertex (grey) or a voxel (black). The unity line of $x = y$ is shown as a reference.
(PDF)

**S3 Fig. Blood perfusion is higher in cortical versus the subcortical tissue.** There exists lower blood perfusion in subcortical voxels compared to the cortical vertices in both male and female participant groups (male: $t = 22.96$, $p_{\text{two-sided}} = 1.25 \times 10^{-96}$; female: $t = 30.65$, $p_{\text{two-sided}} = 3.96 \times 10^{-159}$). Each dot corresponds to the mean cortical perfusion (grey) or mean subcortical perfusion per participant (black).
(PDF)

**S4 Fig. Cortical blood perfusion participant loadings and the score map relevance to functional connectivity (FC) strength.** (a) First principal component loadings are shown per participant (male: blue, female: red). PC loadings assess the similarity between individual participants' cerebral blood perfusion pattern and the composite map captured by the PC.

(b) Individual-level Pearson correlation values between each participant's perfusion map and their FC strength map (male: blue, female: red). Here, FC strength for a region is defined as the absolute weighted sum of all edges connected to that region. In the developmental window (5–22 years), coupling between FC strength and regional perfusion develops ($\rho = 0.09$, $p_{\text{permutation}} = 2.60 \times 10^{-2}$, 1 000 repetitions); and in the aging window (36–100 years) decoupling between the two happens ($\rho = -0.24$, $p_{\text{permutation}} = 9.99 \times 10^{-4}$, 1 000 repetitions). (c) Correlation between average cortical perfusion ($y$-axis) and average absolute functional connectivity strength ($x$-axis) ($r = 0.26$). The significance of the correlation is assessed via permutation testing ($p_{\text{permutation}} = 9.99 \times 10^{-4}$, 1 000 repetitions). Each dot represents data from an individual participant, with colors corresponding to their age.
(PDF)

**S5 Fig. Comparison of blood perfusion maps from PET and ASL imaging.** (a) Cerebral blood flow map from PET imaging is shown on the inflated and 2D flat cortical surfaces (fsLR). (b) Correlation between the $z$-scored cerebral blood perfusion from PET data ($x$-axis) and cerebral blood perfusion score from ASL imaging ($y$-axis) ($r = 0.63$; $p_{\text{spin}} = 9.99 \times 10^{-4}$). Inferior temporal cortex has low blood perfusion in both PET and ASL data, indicating that the low perfusion in this region is not caused by ASL modality-specific signal drop-out.
(PDF)

**S6 Fig. First principal component of cerebral blood perfusion data after regressing out linear and non-linear effects of sex, age and sex-age interactions.** We first regress out the linear and non-linear effects of age, sex and their interactions from participants' cerebral blood perfusion maps using general linear models with the following equation: perfusion = $\beta_0 + \beta_1 \times \text{age} + \beta_2 \times \text{sex} + \beta_3 \times \text{sex} \times \text{age} + \beta_4 \times \text{age}^2 + \beta_5 \times \text{sex} \times \text{age}^2 + \beta_6 \times \text{age}^3 + \beta_7 \times \text{sex} \times \text{age}^3$. Next, we concatenate the cleaned individual data, $z$-score them and perform PCA on the concatenated data matrix. The first component explains about 72.8% of the variance in the data. (a) The brain score map of the first principal component is shown on lateral and medial views of the inflated and 2D flat cortical surfaces (fsLR); the volumetric part is shown on the sagittal view of the T2-weighted group-average template (MNI152). (b) Participant-specific loadings for PC1 are also shown. (c) The obtained brain score map is highly consistent with the PC map in Fig 2A and 2B. The correlation of the obtained score map (after regressing out the co-variates) with the first score perfusion map (shown in Fig 2A and 2B) is equal to 0.93.
(PDF)

**S7 Fig. Quantification of the first perfusion score map at subcortical level.** (a) Mean score value per Tian-S4 subcortical parcel is shown here. Values are sorted from high to low perfusion scores (low perfusion score indicates less cerebral blood flow). (b) The Tian-S4 parcellation is shown on a MNI152 T1-weighted volume space (sagittal and transverse views) [87]; see S3 Table for full parcel names.
(PDF)

**S8 Fig. Relating perfusion and laminar differentiation.** The top row shows the average $z$-scored transcriptomic signature of cortical layer I–III, IV, and V–VI, respectively [88]. The bottom row presents scatter plots with perfusion scores on the $y$-axis and the average $z$-score of gene maps for each layer on the $x$-axis. For layer I–III, the transcriptomic signature is defined using 13 genes (C1QL2, C20orf103, CARTPT, DISC1, GLRA3, GSG1L, IGSF11, INPP4B, MFGE8, PVRL3, RASGRF2, SV2C, and WFS1). However, three of these genes (C20orf103, PVRL3, and DISC1) do not meet the differential stability of 0.1 and hence are excluded from the analysis. The transcriptomic signature for layer IV is based on 5 genes (COL6A1, CUX2, TRMT9B, GRIK4, and RORB), all of which pass the differential stability

threshold of 0.1. For layer V–VI, there are 28 gene markers (ADRA2A, AKR1C3, ANXA1, B3GALT2, CDH24, CTGF, ETV1, FAM3C, FOXP2, HTR2C, KCNK2, NPY2R, NR4A2, NTNG2, OPRK1, PCDH17, PCDH20, PCP4, PDE1A, RPRM, RXFP1, SNTB1, SYT10, SYT6, TLE4, TOX, TRIB2, and VAT1L). Four of these genes (CDH24, FAM3C, NPY2R, and TRIB2) do not meet the differential stability of 0.1 and are excluded from the analysis.
(PDF)

**S9 Fig. Relating perfusion to molecular signaling systems and laminar organization.** We use dominance analysis to assess the contribution of biological features to regional variations in cerebral blood perfusion. Dominance analysis distributes the model's explained variance among the input variables, allowing comparison of their relative contribution to predicting perfusion patterns. (a, b) show results for a model including neurotransmitters, neuropeptides and cortical laminar profiles; and (c, d) show results for a model including only neurotransmitters and neuropeptides. Model fit, expressed as adjusted $R^2$, reaches to 0.64 for the first model (a) and to 0.59 for the second model (c). Asterisks denote significant model fits (in both cases: $p_{spin} = 9.99 \times 10^{-3}$, $n_{spin} = 1\,000$). The percent contributions of each input variable normalized by the total model fit is shown in (b, d).
(PDF)

**S10 Fig. Sex differences in developmental perfusion trajectories.** To test the interaction between age and sex on cerebral perfusion during development, we fit a linear model including main effects of age, sex, and their interaction: perfusion = $\beta_0 + \beta_1 \times$ age $+ \beta_2 \times$ sex $+ \beta_3 \times$ age $\times$ sex (male = 1, female = 0; $\beta_0 = 112.44$). The model reveals age-related decline in perfusion ($\beta_1 = -1.81$, $p = 2.4 \times 10^{-27}$). There is also a sex difference in baseline perfusion ($\beta_2 = 8.42$, $p = 0.026$), and an interaction between age and sex ($\beta_3 = -1.05$, $p = 3.26 \times 10^{-5}$), suggesting that males exhibit a steeper age-related decline in grayordinates' perfusion compared to females (male: blue, female: red).
(PDF)

**S11 Fig. Mean age-effect in subcortical regions.** Average age-effect per parcel is shown for HCP-D (a) and HCP-A (b) datasets. (c) Subcortical parcels are defined based on Tian-S4 subcortical parcellation [87]. See S3 Table for full parcel names.
(PDF)

**S12 Fig. Age and sex-effects of cerebral blood perfusion in the HCP-D dataset.** Top: The general linear model (GLM) coefficients for age and biological sex (male = 1, female = 0) are shown on the cortical fsLR surface maps and MNI152 T2-weighted volume space. Bottom: Areas with Bonferroni-corrected $p$-values below 0.05 are marked in red.
(PDF)

**S13 Fig. Spearman correlation between cerebral blood perfusion and age in the HCP-D dataset.**
(PDF)

**S14 Fig. GAMLSS trajectories to model blood perfusion (versus age) in development.** (a) Each dot corresponds to a participant's mean brain blood perfusion level in the HCP-D cohort (male: blue, female: red). A separate GAMLSS model is fitted for each sex-group to capture age-related changes in grayordinates' perfusion during development. (b) Parcel-wise trajectories of cerebral blood perfusion across age are modeled using GAMLSS in each of the 400 Schaefer parcels. Each panel displays the fitted trajectories for male and female participants. (c) Model fit values are shown on lateral and medial views of the inflated and 2D flat cortical surfaces (fsLR). Higher values indicate better fits. (d) Comparison of mean model

fit values across seven canonical intrinsic functional networks introduced by Yeo et al. [124] shows that transmodal regions exhibit higher model fits than unimodal regions. Additionally, model fits are higher in males than in females.
(PDF)

**S15 Fig. Sex-stratified trajectories of perfusion during development.** Fitted GAMLSS trajectories of cerebral perfusion are shown for each of the 400 Schaefer parcels, grouped by seven canonical intrinsic functional networks introduced by Yeo et al. [124].
(PDF)

**S16 Fig. Age and sex-effects of cerebral blood perfusion in the HCP-A dataset.** Top: The general linear model (GLM) coefficients for age and biological sex (male = 1, female = 0) are shown on cortical fsLR surface maps and MNI152 T2-weighted volume space. Bottom: Areas with Bonferroni-corrected $p$-values below 0.05 are marked in red.
(PDF)

**S17 Fig. Spearman correlation between cerebral blood perfusion and age in the HCP-A dataset.**
(PDF)

**S18 Fig. GAMLSS trajectories to model blood perfusion (versus age) in aging.** (a) Each dot corresponds to a participant's mean brain blood perfusion level in the HCP-A cohort (male: blue, female: red). A separate GAMLSS model is fitted for each sex-group to capture age-related changes in grayordinates' perfusion during aging. (b) Parcel-wise trajectories of cerebral blood perfusion across age are modeled using GAMLSS in each of the 400 Schaefer parcels. Model fit values are shown on lateral and medial views of the inflated and 2D flat cortical surfaces (fsLR). Higher values indicate better fits.
(PDF)

**S19 Fig. First principal component of arterial transit time across HCP-D and HCP-A participants.** (a) To generate a representative arterial transit time (ATT) map, the vertex/voxel-wise ATT map of participants are $z$-scored and then combined into a single data matrix. Principal component analysis (PCA) is applied to derive the first principal component (PC1; explaining 22.9% of the variance). The first PC score map is shown on lateral and medial views of the inflated and 2D flat cortical surfaces (fsLR). The volumetric part of the map is shown on the sagittal view of the T2-weighted group-average template (MNI152). Here positive values mark regions with late blood arrival times. The second principal component explains 4.07% of the variance in the data. (b) Loadings of the first principal component are shown per participant (male: blue, female: red).
(PDF)

**S20 Fig. Biomarkers versus age.** Relationship between the raw biomarkers used in the PLS analysis ($y$-axis) and participants' age ($x$-axis) (male: blue, female: red).
(PDF)

**S21 Fig. The $z$-scored weight distribution of biomarkers.** PLS bootstrapping is used to assess the robustness and stability of PLS weights [291,292].
(PDF)

**S22 Fig. Relating blood perfusion and biomarkers after regressing out the linear effect of age and sex.** (a) Using a GLM model we regress out the linear effect of age and sex from both the biomarker data and the vertex/voxel-wise perfusion data, we parcellate the perfusion maps using Schaefer-400 atlas and relate the biomarkers (excluding age and sex) to blood perfusion

maps of participants using the partial least squares (PLS) analysis. The first latent variable accounts for 94.30% of the covariance between cortical blood perfusion and biomarker profiles of HCP-A participants ($p = 9.99 \times 10^{-4}$, 1 000 repetitions). (b) The bar plot visualizes the contribution (effect size) of individual biomarkers to the first latent variable. The significance of each biomarker's contribution to the overall pattern is assessed by bootstrap resampling (1 000 repetitions). (c) The brain loadings of the first latent variable are shown on the inflated and 2D flat cortical surfaces (fsLR).
(PDF)

**S23 Fig. Cerebral blood perfusion covariance matrix gradients in subcortex.** The S4 Tian parcels are used to parcellate the gradient values in subcortex [87]. See S3 Table for full parcel names. S = superior; A = anterior; P = posterior; I = inferior; R = right; L = left.
(PDF)

**S1 Table. Multi-modal Glasser parcellation table.**
(XLSX)

**S2 Table. Mean blood perfusion score per cortical parcel.** Parcels are defined based on the multi-modal Glasser parcellation of cerebral cortex, which includes 180 parcels in each hemisphere [86]. For each parcel, we calculate the mean perfusion scores across both hemispheres. This approach ensures that we have a single value per parcel, rather than separate values for the left and right hemispheres. Fig 2A shows a detailed map with areal borders overlaid on the perfusion map. Full name of parcels is provided in S1 Table.
(PDF)

**S3 Table. Nomenclature for Tian-S4 subcortical functional parcellation [87].**
(PDF)

**S4 Table. GO table.**
(XLSX)

**S5 Table. Cell type table.**
(XLSX)

**S6 Table. Neurotransmitter receptors.** Values in parentheses (under #Participants) indicate number of females.
(PDF)

**S7 Table. Comparison of male and female cerebral blood perfusion values in different age bins.** The *t*-statistics and *p*-values per each age-bin are shown. "#" denotes the number of participants.
(PDF)

**S8 Table. Neurosynth terms.** List of 124 used Neurosynth terms in this study.
(PDF)

## Author contributions

**Conceptualization:** Asa Farahani, Bratislav Misic.

**Data curation:** Asa Farahani, Zhen-Qi Liu.

**Formal analysis:** Asa Farahani.

**Funding acquisition:** Bratislav Misic.

**Investigation:** Asa Farahani, Zhen-Qi Liu, Bratislav Misic.

**Methodology:** Asa Farahani, Zhen-Qi Liu, Eric G. Ceballos, Justine Y. Hansen, Karl Wennberg, Bratislav Misic.

**Project administration:** Bratislav Misic.

**Resources:** Zhen-Qi Liu, Eric G. Ceballos, Justine Y. Hansen, Karl Wennberg, Bratislav Misic.

**Supervision:** Yashar Zeighami, Mahsa Dadar, Claudine J. Gauthier, Alain Dagher, Bratislav Misic.

**Validation:** Zhen-Qi Liu, Yashar Zeighami, Mahsa Dadar, Claudine J. Gauthier, Alain Dagher, Bratislav Misic.

**Visualization:** Asa Farahani, Zhen-Qi Liu, Bratislav Misic.

**Writing – original draft:** Asa Farahani, Bratislav Misic.

**Writing – review & editing:** Asa Farahani, Zhen-Qi Liu, Eric G. Ceballos, Justine Y. Hansen, Karl Wennberg, Yashar Zeighami, Mahsa Dadar, Claudine J. Gauthier, Alain Dagher, Bratislav Misic.

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
