## [Editor Report · Decision Letter 0]

17 Mar 2025

Dear Dr Misic,

Thank you for submitting your manuscript entitled "Cerebral blood perfusion across biological systems and the human lifespan" for consideration as a Research Article by PLOS Biology.

Your manuscript has now been evaluated by the PLOS Biology editorial staff, as well as by an academic editor with relevant expertise, and I am writing to let you know that we would like to send your submission out for external peer review.

Once your full submission is complete, your paper will undergo a series of checks in preparation for peer review. After your manuscript has passed the checks it will be sent out for review. To provide the metadata for your submission, please Login to Editorial Manager (https://www.editorialmanager.com/pbiology) within two working days, i.e. by Mar 19 2025 11:59PM.

Kind regards,

Taylor Hart, on behalf of Lucas Smith while he is out of office

Taylor Hart, PhD

Associate Editor

thart@plos.org

-----

Lucas

Lucas Smith, Ph.D.

Senior Editor

PLOS Biology

lsmith@plos.org

---

## [Decision Letter · Decision Letter 1]

29 Apr 2025

Dear Dr Misic,

Thank you for your patience while your manuscript "Cerebral blood perfusion across biological systems and the human lifespan" was peer-reviewed at PLOS Biology. It has now been evaluated by the PLOS Biology editors, an Academic Editor with relevant expertise, and by several independent reviewers. In light of the reviews, which you will find at the end of this email, we would like to invite you to revise the work to thoroughly address the reviewers' reports.

As you will see below, the reviewers report that the study is generally rigorous and offers interesting insights, but they have a number of suggestions to strengthen the study further. We think that these should be addressed before we can consider the paper further for publication. We note that reviewer 3 has suggested a number of additional analyses which we think would improve the study, and so we encourage you to add the requested data, if feasible. We would particularly encourage you to add the age effect request in Reviewer 3's point 2. On the other hand, we are OK with the format of the results section, and would not require that you remove methodological detail from the section as requested by Reviewer 3 (point 3).

Given the extent of revision needed, we cannot make a decision about publication until we have seen the revised manuscript and your response to the reviewers' comments. Your revised manuscript may be sent for further evaluation by all or a subset of the reviewers.

**IMPORTANT - SUBMITTING YOUR REVISION**

*Re-submission Checklist*

*Published Peer Review*

*PLOS Data Policy*

*Blot and Gel Data Policy*

Sincerely,

Luke

Lucas Smith, Ph.D.

Senior Editor

PLOS Biology

lsmith@plos.org

REVIEWS:

Reviewer #1, Danny JJ Wang (note, reviewer 1 has signed this review): This is a revision, the authors have been responsive to previous review, their methods are quite rigorous. I only have a few minor comments:

1) Have you tried to look at the differences in the distribution and magnitude of perfusions in white matter between male and female? Because many studies have found gender differences in brain white matter, I am curious whether perfusion in white matter can reflect certain properties?

2) There is no description about whether the correlation between perfusion and gene expression is based on ROI or vertex? We know that the size of each ROI is different. AHBA map should be based on ROI. If you use the perfusion score map calculated based on vertex to do the correlation, will it be biased?

Reviewer #2, Peter Zhukovsky (note, reviewer 2 has signed this review): The authors leverage HCP aging and development data (n>1000) across lifespan to provide a comprehensive analysis of cerebral blood perfusion. They leverage brain map-to-brain map associations to show that higher levels of perfusion are seen in regions with high metabolic demand and resting-state functional hubs. Using imaging transcriptomics with AHBA, they show that higher perfusion is seen in the granular layer IV, and correlative evidence suggests its related to GO pathways such as vasculogenesis and potassium and sodium transport and glucose. Interesting age and sex effects and associations with blood biomarkers are found. Sex effects appear to emerge around adolescence; common plasma proteins explain a lot of variance in the perfusion map in males and females separately. Overall, a lot of work and analyses have been completed for this project, with comprehensive visualizations and supplementary information. The link to neurodegenerative disorders would have been stronger if a dementia dataset with CBF/perfusion was included, ie analyzing individual CB perfusion maps from patients with varying levels of cognitive impairment, but it does not detract from the interesting insights into the anatomical patterns of perfusion distribution across the brain. The PLS analysis is one of the strongest aspects of this paper in my view. The implications of the tight coupling between common plasma biosignatures and CBF perfusion should be discussed more. I have several specific points below, mostly related to figures and minor issues, with one question on statistical reporting for Figure 4. While the manuscript is well written and structured overall, the writing style could be tightened up at times, e.g. rhetorical questions in the results or using 'Zooming in' in the discussion could be compressed for a more succinct writing style.

Fig 4b the 2d map may not be necessary here given that its hard to make out which region is which on the 2d map - just keeping a larger version of the 3d brain maps would be better instead

Blood perfusion across development section - the authors need to report main effects vs interactions for age, sex and age x sex. The way the section is described right now suggests an age x sex interaction, whereby up to age 15 or so there are no sex differences, which then emerge and get stronger with age. That makes sense; however this conclusion should be supported by appropriate stats (ie interaction testing).

"Which brain regions experience the greatest developmental changes in blood perfusion?" - results should not include rhetorical questions

Figure 5 - there is a lot of white space; the brain image scaling and p-spin/atlas references could be improved slightly

Figure 5d - please don't call plasma proteins 'physiological measure', it's too general and is more commonly used to refer to heart rate and breathing monitoring during an fmri scan. Either plasma proteins or plasma biomarkers are much more appropriate here.

Figure 5d - female estradiol is a key marker for the 'male' group? Seems contradictory, could the authors please comment?

Figure 7 and the associated analysis could be moved to supplementary if needed

Figure S3c - strength is misspelled

Figure S4b - "comparison … <delete coming> from PET and ASL" for stylistic improvements

Figure S13 - also please update physiological measures here

For future PLS - if the authors are using pls bootstrap for robustness of weights, its worth reporting the weights as Z-scores to assess stability (see Morgan et al 2019 PNAS, Zhukovsky et al 2022 PNAS for code). Is it a PLS correlation or a PLS-regression? PLSC/PLSR makes this clear to the reader.

Reviewer #3: This work analyzed pseudo-continuous arterial spin labeling (ASL) data from the HCP Lifespan studies (aged 5-22 years, and 36-100 years) to reconstruct a high-resolution normative cerebral blood perfusion map. The authors investigate how blood perfusion co-localizes with micro- and mesoscale features, including neurotransmitter signaling, cell types, and cortical layers. Then, the authors investigate how blood perfusion changes through the lifespan and in multiple neurodegenerative diseases. Finally, the authors show that interregional coordination of blood perfusion can be used to map arterial territories.

Overall, this work is very interesting and holds significant value for the field of neuroscience and research on brain development. However, I do have several major and minor concerns/issues, which are itemized below.

1. Cerebral blood perfusion is dynamic and changes throughout the lifespan. To quantify the effect of age on cerebral blood perfusion during development, the authors constructed linear regression models. However, the development process is not always linear (Bethlehem Richard AI, et al. Nature 2022). A generalized additive model for location, scale, and shape (GALSS), a robust method for modeling nonlinear growth trajectories, would be useful for constructing the development process across the human lifespan.

2. The physical meaning of age coefficients (beta) of age-related changes is difficult to comprehend. To provide insight into the overall magnitude and direction of regional age effects, many previous studies (such as Sydnor Valerie, et al. Nature neuroscience, 2023; Li Jiao, et al. Plos Biology, 2024; Lianglong Sun, et al. Nature neuroscience, 2025) have represented the age effects (R2 values) as the age-related regional changes during development. Therefore, I suggest replacing the t-statistic with the age effect (R2 values).

3. The Results section contains too much description of the Methods. The authors should mainly describe the results and what they reveal.

4. The authors used ALS data from the HCP-D and HCP-A datasets to investigate age-related changes in cerebral blood perfusion. However, data from individuals aged 23 to 35 years are missing. Therefore, the use of the term "lifespan" in the title is misleading and not appropriate. The term "HCP Lifespan studies (5-100 years)" should be revised to reflect the actual age coverage of the datasets used.

5. The authors estimate the cortical Layer IV pattern using the average expression map of five genes preferentially expressed in the human granular layer IV. However, the rationale for focusing exclusively on Layer IV-specific gene expression remains unclear. Prior work (e.g., Burt et al., 2018) has demonstrated layer-specific gene expression across three broader laminar groups: Layers 1-3, Layer 4, and Layers 5-6. To strengthen the biological interpretability of the findings, I recommend that the authors provide a justification for the selective focus on Layer IV, and, if feasible, consider incorporating gene expression patterns from additional cortical layers to offer a more comprehensive characterization of laminar contributions.

6. The authors utilize two cortical parcellation atlases—the Glasser and Schaefer-400 functional atlas. However, the rationale for using the Glasser atlas to quantify cortical perfusion scores, while employing the Schaefer-400 atlas for subsequent spatial association analyses remains unclear. This inconsistency may affect the interpretability and comparability of results across different parts of the study. I recommend that the authors clarify their reasoning for choosing different atlases and, if feasible, consider using the same atlas consistently throughout the analyses to enhance methodological coherence.

7. The authors assess the associations between cortical perfusion scores and micro- and macros-scale features (such as oxygen metabolism, laminar differentiation, functional connectivity strength, gene expression, cell types, neurotransmitter receptors, and neuropeptide receptors). However, it is unclear whether subcortical structures (e.g., those defined by the Tian functional atlas) were excluded from these analyses.

8. It would be interesting to explore the relative contributions of these micro- and macro-scale features to cortical perfusion scores beyond simple spatial correlations—for example, through multivariate modeling and dominance analysis.

9. The authors employed a non-parametric "spin test" to assess spatial correlations. However, this method is specifically designed for cortical surface data and does not accommodate subcortical regions. Given that the current analyses include cortical and subcortical areas, the spin test may not be the most suitable approach. Additionally, the spin test is limited by potential biases introduced when the medial wall—typically lacking data—is rotated into cortical areas (Markello et al., NeuroImage, 2021). An alternative method, such as variogram matching (Burt et al., NeuroImage, 2020), which can handle both cortical and subcortical regions and account for spatial autocorrelation more comprehensively, would be more appropriate for the current study.

---

## [Decision Letter · Decision Letter 2]

12 Jun 2025

Dear Dr Misic,

Thank you for your patience while we considered your revised manuscript "Cerebral blood perfusion across development and aging: Links with multi-scale organization of the human brain" for publication as a Research Article at PLOS Biology. This revised version of your manuscript has been evaluated by the PLOS Biology editors, the Academic Editor and by two of the original reviewers.

As you will see in their comments, below, both of the reviewers are satisfied by the revision. However, before we can editorially accept your study we need you to address a few data and other policy-related requests in another short revision. These are detailed below.

**EDITORIAL REQUESTS** Please address the following points.

1) TITLE: We would like to propose a tweak to the title, to make it a bit more streamlined. If you agree, we suggest you change it to something like:

"Mapping cerebral blood perfusion and its links to multi-scale brain organization across the human lifespan"

or possibly

"Cerebral blood perfusion is linked to functional and structural organization across the human lifespan in health and disease"

2) ETHICS STATEMENT: I see your ethics statement currently reads "All study procedures of the HCP protocol are approved by the Institutional Review Board at Washington University in St. Louis." Are you able to also provide the approval number for the protocol that was approved by WUSTL IRB? And can you confirm that this study adhered to the principles expressed in the Declaration of Helsinki?

3) DATA: Thank you for providing the underlying data and code on github. While this is fine, please note that we cannot accept sole deposition of code in GitHub, as this could be changed after publication. We therefore ask that you can archive this version of your publicly available GitHub code to Zenodo. Once you do this, it will generate a DOI number, which you will need to provide in the Data Accessibility Statement (you are welcome to also provide the GitHub access information). See the process for doing this here: https://docs.github.com/en/repositories/archiving-a-github-repository/referencing-and-citing-content

4) DATA: I also noticed on your GitHub page, you mention that the 'Arterial Spin Labeling (ASL) dataset used in this study is provided by the HCP Lifespan studies, and cannot be publicly released by us. For more details and to request access to the dataset, please visit the HCP Lifespan website.'

I do think that this is an allowable data restriction under our policy - but we ask that you add a similar note to your data availability statement, in our online system. You should also include the URL of the website where researchers can gain access to this data.

We expect to receive your revised manuscript within two weeks.

*Published Peer Review History*

*Press*

Sincerely,

Luke

Lucas Smith, Ph.D.

Senior Editor

lsmith@plos.org

PLOS Biology

Reviewer remarks:

Reviewer #2, Peter Zhukovsky (note, reviewer 2 has signed this review): The authors addressed all my comments. The manuscript included a plethora of advanced analyses to begin with, and the authors extensively revised the manuscript with regard to the points raised. I believe it will make a valuable addition to the field of neuroscience.

Reviewer #3, Wei Liao (note, reviewer 3 has signed this review): The authors have addressed all issues. No more questions.

---

## [Editor Report · Decision Letter 3]

24 Jun 2025

Dear Dr Misic,

Thank you for the submission of your revised Research Article "Mapping cerebral blood perfusion and its links to multi-scale brain organization across the human lifespan" for publication in PLOS Biology and thank you for addressing our editorial requests in this revision. On behalf of my colleagues and the Academic Editor, Laura D. Lewis, I am pleased to say that we can in principle accept your manuscript for publication, provided you address any remaining formatting and reporting issues. These will be detailed in an email you should receive within 2-3 business days from our colleagues in the journal operations team; no action is required from you until then. Please note that we will not be able to formally accept your manuscript and schedule it for publication until you have completed any requested changes.

**IMPORTANT - Two last editorial notes:

1) Thank you for confirming that the human studies were conducted in accordance with the principles expressed in the declaration of Helsinki. Can you please update the relevant section of your methods section (where you discuss the IRBs) to reflect that.

2) Thank you for updating the data availability statement in your manuscript to include more details about accessing the HCP datasets. I have taken the liberty of updating your 'data availability statement' in our online system to reflect these details, as I believe that is the version that will be published with your paper. It currently reads:

All code used to perform the analyses are available on GitHub at https://github.com/netneurolab/Farahani_Blood_Perfusion/ and on Zenodo at https://zenodo.org/records/15708107/ (DOI: 10.5281/zenodo.15708107). Arterial spin labeling (ASL), functional MRI, and structural MRI are accessible through the Human Connectome Project–Development (HCPD; https://www.humanconnectome.org/study/hcp-lifespan-development/) and the Human Connectome Project–Aging (HCP-A; https://www.humanconnectome.org/study/hcp-lifespan-aging/)

Please take a minute to log into Editorial Manager at http://www.editorialmanager.com/pbiology/, click the "Update My Information" link at the top of the page, and update your user information to ensure an efficient production process. Please also be sure to look through the data availability statement in our online system, and make sure it looks good to you.

PRESS

Sincerely, 

Luke

Lucas Smith, Ph.D.

Senior Editor

PLOS Biology

lsmith@plos.org